# Towards the Training of Deeper Predictive Coding Neural Networks

Chang Qi[1]  Matteo Forasassi[1]  Thomas Lukasiewicz[1 2]  Tommaso Salvatori[1]

## Abstract

Predictive coding networks are neural models that perform inference through an iterative energy minimization process, whose operations are local in space and time. While effective in shallow architectures, they suffer significant performance degradation beyond five to seven layers. In this work, we show that this degradation is caused by exponentially imbalanced errors between layers during weight updates, and by predictions from the previous layers not being effective in guiding updates in deeper layers. Furthermore, when training models with skip connections, the energy propagated by the residuals reaches higher layers faster than that propagated by the main pathway, affecting test accuracy. We address the first issue by introducing a novel precision-weighted optimization of latent variables that balances error distributions during the relaxation phase, the second issue by proposing a novel weight update mechanism that reduces error accumulation in deeper layers, and the third one by using auxiliary neurons that slow down the propagation of the energy in the residual connections. Empirically, our methods achieve performance comparable to backpropagation on deep models such as ResNets, opening new possibilities for predictive coding in complex tasks.

## 1. Introduction

Training deep learning models is extremely expensive in terms of energy consumption. To address this problem, a recent direction of research is studying the use of alternative accelerators that leverage the properties of physical systems to perform computations (Wright et al., 2022; Momeni et al., 2025), such as in-memory computations using memristor crossbars (Tsai et al., 2018; Haensch et al., 2018). However,

transitioning to new hardware without altering the main algorithm — error backpropagation (Rumelhart et al., 1986) — has proven to be challenging due to two central issues: the requirement for sequential forward and backward passes, and the need to analytically compute gradients of a global cost function. These issues do not arise when using learning algorithms that rely on computations that are local in space and time (i.e., all updates rely on locally available information only) (Bengio, 2014; Hinton, 2022). A popular example is equilibrium propagation, a framework that allows for the learning of the parameters of a neural network by simulating a physical system brought to an equilibrium (Scellier & Bengio, 2017). This physical system is usually defined via an energy function that describes the state of a neural network in terms of its weights and neurons, with different functions describing different systems (Krotov & Hopfield, 2016).

In recent years, researchers have devoted significant effort to scaling up the deployment of energy-based learning algorithms. Two recent works have benchmarked multiple variants of common learning algorithms using the Hopfield energy function (Scellier et al., 2024) and the predictive coding energy (Pinchetti et al., 2025), showing that this class of models can match the performance of standard deep learning when training relatively shallow networks of up to five or seven layers. However, this success does not extend to deeper architectures, where performance degrades substantially. Notably, the degradation is even more severe in predictive coding implementations of residual networks, which perform worse than equally deep models without skip connections. Since the major advances of modern deep learning rely on very deep architectures with residual connections, understanding and overcoming these limitations is a critical step toward scaling predictive coding networks to the regimes where deep learning has been most successful.

To understand the poor scalability of predictive coding networks, it is instructive to examine how energy propagates across depth. It has recently been shown that in a three-layer model, the energy concentrated in a layer can be up to an order of magnitude larger than the energy concentrated in the layer before (Pinchetti et al., 2025). While such shallow models can still achieve good test accuracies, we conjecture that this 'energy imbalance' becomes a critical bottleneck in deeper architectures, leading to performance degrada-

[1]Vienna University of Technology, Austria [2]University of Oxford, UK. Correspondence to: Tommaso Salvatori <tommaso.salvatori@verses.ai>.

*Proceedings of the 43rd International Conference on Machine Learning*, Seoul, South Korea. PMLR 306, 2026. Copyright 2026 by the author(s).

tion in a way that is conceptually similar to the *vanishing gradient* problem (Hochreiter, 1998). More precisely, this imbalance prevents the effective propagation of energy — and crucially, the associated error information — from the output layer back to the early layers, creating two problems: first, it prevents the model from fully leveraging its depth, as early layers receive insufficient error signals for effective training; second, latent states may diverge substantially from their forward pass values due to the excessive energy in later layers.

Addressing this energy imbalance requires mechanisms that can adaptively modulate the relative influence of different layers as errors propagate through the network, a challenge similar to what biological neural systems solve through precision-based regulation. In predictive processing theories, precision refers to the estimated reliability of a prediction error (formally, its inverse variance), and it is thought to be dynamically regulated to balance bottom-up and top-down signals across cortical hierarchies (Feldman & Friston, 2010; Bastos et al., 2012). Despite its central role in biological inference, most machine learning formulations of predictive coding set precision to 1 for simplicity (Whittington & Bogacz, 2017), thereby overlooking its potential as a powerful mechanism to stabilize learning dynamics. The first contribution of this work is to propose leveraging precision weighting to regularize energy propagation in predictive coding networks.

We begin by analyzing the energy propagation of deep convolutional architectures for both predictive coding (PC) and incremental PC (iPC) — a recently introduced variant of PC that updates weights and neurons simultaneously (Salvatori et al., 2024). Building on the insights from this analysis, we propose time-dependent precisions that address the identified issues. Our results show that this substantially improves test accuracy, supporting our conjecture of a causal link between energy propagation and empirical performance. More broadly, we propose two algorithmic improvements (spiking precision and a novel weight-update mechanism) and two architecture adaptation (PC-tailored batch normalization and auxiliary neurons for skip connections) that enable PC to achieve competitive performance with backprop on image classification benchmarks[1], including VGG models with up to 15 layers and ResNets18 on Tiny ImageNet. Our contributions can be detailed as follows:

- We show that in models trained with PC, the energy is orders of magnitude larger in layers closer to the output, supporting the hypothesis that information fails to propagate effectively to the earlier layers. This phenomenon is less pronounced in iPC, where continuous weight updates rapidly reduce the excess energy. However, these updates

[1]The code can be found at: https://github.com/changqi97/DeepPCNs.

result in even lower test accuracy.

- To mitigate this imbalance, we propose dynamical precision-weightings that depend on both time and layer depth. The most effective variant, which we call *spiking precision*, applies very large precisions as soon as the energy reaches a given layer, thereby boosting it forward. Experiments show that this method regulates the energy imbalance and improves test accuracy in deep PC models. In the case of iPC, spiking precisions alone are already sufficient to achieve performance comparable to backpropagation in deep networks.

- To further improve the performance of standard PC models, we introduce a novel weight-update mechanism that modifies how parameters are updated. This method combines predictions computed at initialization (hence adding a degree of implausibility, as they have to be stored in memory) with the neural activities at convergence, resulting in more effective updates. With this approach, PC attains performance on par with backpropagation and iPC on deep convolutional models. In addition, we propose a variant of batch normalization (Ioffe & Szegedy, 2015) tailored for PC, which further enhances performance.

- While effective for VGG-like models, when training PC-based ResNets (He et al., 2016), we still observe a significant drop in performance. We conjecture that this is caused by the energy propagated by the residuals, which reach higher layers faster than that propagated by the main pathway, disrupting learning dynamics. We show that this can be addressed by adding extra families of neurons inside the skip connections, that have the sole goal of slowing down the feedback signal of the skip connections so that it reaches the higher layers at the same time as the main one. The results show that such auxiliary neurons allow models trained with PC and iPC to reach performance comparable to these of backprop on ResNet18. These contributions are organized along functional lines: Algorithmic Contributions (Spiking Precision and Forward Update) modify the learning rules, while Architecture Adaptations (Auxiliary Neurons and BatchNorm Freezing) address structural incompatibilities between PC and standard deep architectures.

## 2. Related Works

**Equilibrium Propagation (EP).** EP is a learning algorithm for supervised learning that is largely inspired by contrastive learning on continuous Hopfield networks (Movellan, 1991). Here, neural activities are updated in two phases: In the first, to minimize an energy function defined on the parameters of the neural network; in the second, to minimize the same energy with the addition of a loss function defined on the labels (Scellier & Bengio, 2017). Interest-

ingly, these two phases allow us to approximate the gradient of the loss function up to arbitrary levels of accuracy using finite difference coefficients (Zucchet & Sacramento, 2022). The consequence is that EP can be seen as a technique that allows the minimization of loss functions using arbitrary physical systems that can be brought to an equilibrium, and it has hence been studied in a large number of domains (Scellier, 2024; Kendall et al., 2020). The state of the art is that EP models are able to match the performance of BPTT (BP Through Time) on models with 5 and 7 hidden layers (Scellier et al., 2024), with the exception of hybrid models, which manage to achieve a good performance on models with 15 layers by alternating blocks of layers trained with BP and blocks trained with EP (Nest & Ernoult, 2024).

**Predictive Coding (PC).** The formulation of PC that we use here was developed to model hierarchical information processing in the brain (Rao & Ballard, 1999; Friston, 2005). Intuitively, this theory states that neurons and synapses at one level of the hierarchy are updated to better predict the activities of the neurons of the layers below, and minimize the *prediction error*. Interestingly, the same algorithm can be used as a training algorithm for deep neural networks (Whittington & Bogacz, 2017), where several similarities with backpropagation were observed (Song et al., 2020; Salvatori et al., 2022). To this end, it has been used in a large number of machine learning tasks, from image generation and classification to natural language processing and associative memory (Sennesh et al., 2024; Salvatori et al., 2023; Pinchetti et al., 2022; Ororbia & Kifer, 2022; Salvatori et al., 2021). Again, the state of the art has been reached by training convolutional models with 5 hidden layers, with performance starting to get worse as soon as we use models with 7 layers (Pinchetti et al., 2025). To this end, a recent interesting work has proposed $\mu PC$, a theoretical framework that allows the training of very deep feedforward models (Innocenti et al., 2026). However, this work does not tackle non-feedforward layers, and does not test on datasets larger than MNIST. The connection between PC and EP is well explained by the concept of bi-level optimization (Zucchet & Sacramento, 2022), where the neural activities used for learning are the equilibrium state of a physical system.

## 3. Background

Let us consider a network with $L$ layers, and let us denote $\mathbf{W}^l$ and $\mathbf{x}_t^l$ the weight parameters and the neural activities of layer $l$, respectively. Note that, differently from standard models, the neural activities are variables of the model, optimized over time steps $t$. This optimization is performed with the goal of allowing the activities of every layer to predict those of the layer below. Together with the neural activities, the two other quantities related to single neurons are the *prediction* $\mu_t^l = \mathbf{W}^l \mathbf{f}\left(\mathbf{x}_t^{l-1}\right)$, given by the layer-wise

operation through an activation function, and the *prediction error*, defined as the deviation of the actual activity from the prediction, that is, $\varepsilon_t^l = \mathbf{x}_t^l - \mu_t^l$. A fourth quantity, usually overlooked in machine learning applications but of vital importance in neuroscience, is the *precision*, or *covariance* $\Sigma_t^l$ of a specific neuron [2]. Differently from the standard literature, we consider the covariance to be time-dependent. Furthermore, instead of updating it to minimize an objective function as done in previous works (Ofner et al., 2021), we will manually define the rule that governs its updates. The predictive coding energy is then the sum of the squared norms of the precision-weighted prediction errors of every layer over time:

$$E_t = \frac{1}{2}\sum_{l=1}^{L}\frac{\|\mathbf{x}_t^l - \mu_t^l\|^2}{\Sigma_t^l} = \frac{1}{2}\sum_{l=1}^{L}\frac{\|\varepsilon_t^l\|^2}{\Sigma_t^l}, \qquad (1)$$

where we consider covariances to be layer-dependent: all the neurons of the same layer will have the same covariance. To this end, we use the same notation when the covariance $\Sigma_t^l$ is a scalar, or a diagonal matrix whose entries are equal to such a scalar.

**Training.** Suppose that we are provided with a labeled data point $(\mathbf{o}, \mathbf{y})$. Training is then performed via a form of bi-level optimization (Zucchet & Sacramento, 2022), divided into three phases. In the first phase, the neural activities of every neuron are initialized via a forward pass, that is, by setting $\mathbf{x}_0^l = \mu_0^l$ for every layer $l$, with $x_0^0 = o$. In the second phase, which we call the *inference* phase, we clamp the output neurons to the label (i.e., we set $\mathbf{x}_L = \mathbf{y}$) and update the neural activities to minimize the total energy. At $t = 0$, this energy is exactly the supervised loss, as all internal errors are zero and only the output mismatch contributes. The update rule is then:

$$\Delta\mathbf{x}_t^l = -\alpha\frac{\partial E_t}{\partial \mathbf{x}_t^l} = \alpha\left(\frac{\varepsilon_t^l}{\Sigma_t^l} - \mathbf{W}^{(l+1)\top}\frac{\varepsilon_t^{l+1}}{\Sigma_t^l}\odot f'(\mathbf{x}_t^l)\right), \quad (2)$$

where $\alpha$ is the learning rate of the neural activities. This phase will continue until it reaches the fixed number of iterations $\mathbf{T}$ or convergence. The third phase is the *learning* phase, where the neural activities $\mathbf{x}_T^l$ are fixed, and the weight parameters are updated to decrease the energy, weighted by a learning rate $\eta$, via the following equation:

$$\Delta\mathbf{W}^l = -\eta\frac{\partial E_T}{\partial \mathbf{W}^l} = -\eta\frac{\varepsilon_T^l f(\mathbf{x}_T^{l-1})}{\Sigma_T^l}, \qquad (3)$$

**Incremental PC (iPC).** An alternative to the bi-level optimization described above is iPC (Salvatori et al., 2024),

---

[2]In predictive coding, and more generally in statistics, the precision matrix is the matrix inverse of the covariance matrix. In this work, we follow the standard convention and divide the prediction error by a factor of $\Sigma$, instead of multiplying it by a factor of $p$.

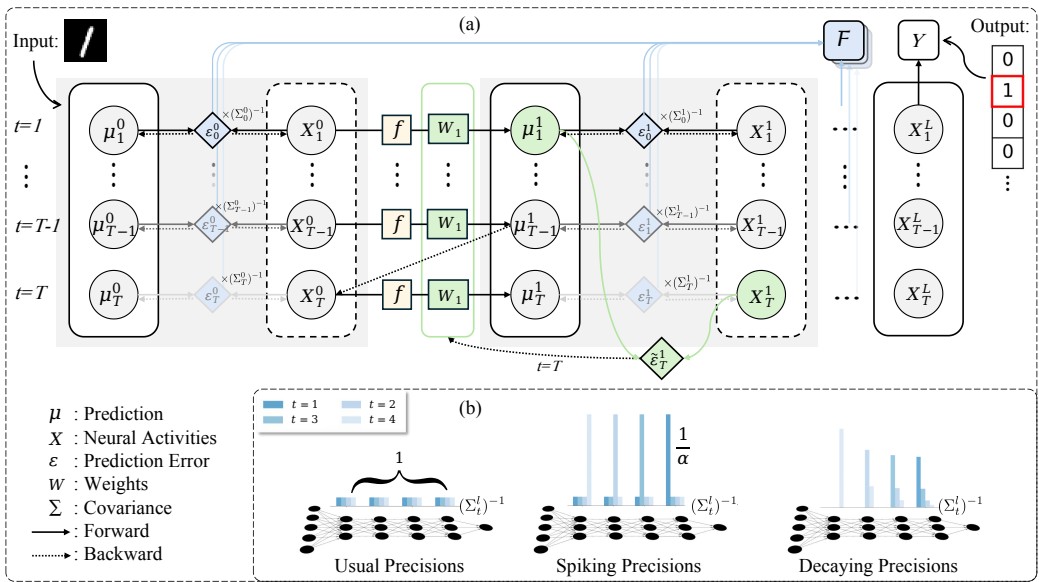

*Figure 1.* (a) Evolution of predictive coding models over multiple time iterations. The green diamond $\tilde{\varepsilon}_T^{l+1}$ refers to the information needed to compute the proposed forward updates. The rest of the figure represents the standard components and mechanisms of a predictive coding network. (b) Visualization of the proposed precision-weighting strategies, where the height of the bar is proportional to the precision at different time steps.

that differs from the standard implementation as also the weight parameters are updated at every time step $t$ until convergence, according to Eq.3. From the variational inference perspective, this schema introduces a form of *incremental expectation-maximization* (Neal & Hinton, 1998), and has been argued to be more biologically plausible than standard PC, as it avoids the need for a control signal that halts the updates of the neural activities and triggers the weight updates. In practice, however, this method has been shown to perform even worse than PC when it is used to train deep models such as VGG7 and VGG9 on CIFAR10 (Pinchetti et al., 2022). We will show that it is this fully automatic training method that will benefit the most from hard-coded precisions.

**Nudging.** Instead of providing the original label $y$ to the model, it is common in the literature to slightly translate the output neurons of the system $\mathbf{x}_0^L$ in the direction of $y$. More precisely, it fixes $\mathbf{x}_t^L = \mu_0^L + \beta(\mathbf{y} - \mu_0^L)$ for every time step $t$, where $\beta$ controls the supervision strength. The sign of $\beta$ determines supervision polarity: positive for standard nudging and negative for inverse supervision. Performing a stochastic sampling from $\{\beta, -\beta\}$ across training epochs and batches is called *center nudging* (Scellier et al., 2024). In practice, PC with centered nudging has been shown to be the best performing method (Pinchetti et al., 2025).

## 4. Methods

In this section, we first study the phenomenon of energy imbalance across different layers, and then use the derived insights to propose time-dependent covariances that address

it. In detail, we propose *spiking precisions*, a method that better distributes the energy across the model by dynamically updating the precisions as soon as individual neurons are reached by the energy term. In practice, we show that this largely improves the performance of models trained with PC and iPC. In the case of iPC, spiking precisions allow us to reach performance comparable to that of backprop. To further boost the performance of standard PC, we introduce a variation of the weight update rule, which leverages neural activities at initialization to perform a better update of the parameters and improve overall model performance. Finally, we address the structural challenges in deep architectures by introducing auxiliary neurons for residual connections and a novel formulation of batch normalization. Figure 1(a) presents a flowchart that intuitively illustrates the modules discussed in subsequent sections, while Figure 1(b) provides a visualization of the covariance matrices.

### 4.1. Motivation: The Energy Imbalance

To study the energy imbalance across different levels of the network, we have tracked the normalized total energy of each layer during training, along with the test and training accuracy, and compared it against that of BP. We have performed a broad study that tests multiple models, datasets, and setups. The detailed plots are reported in Figure. 2. As BP does not have a proper definition of energy, we have used the squared error of every layer computed during the backward pass, equivalent to the error of $PC$ when it comes to the update of the weight parameters.

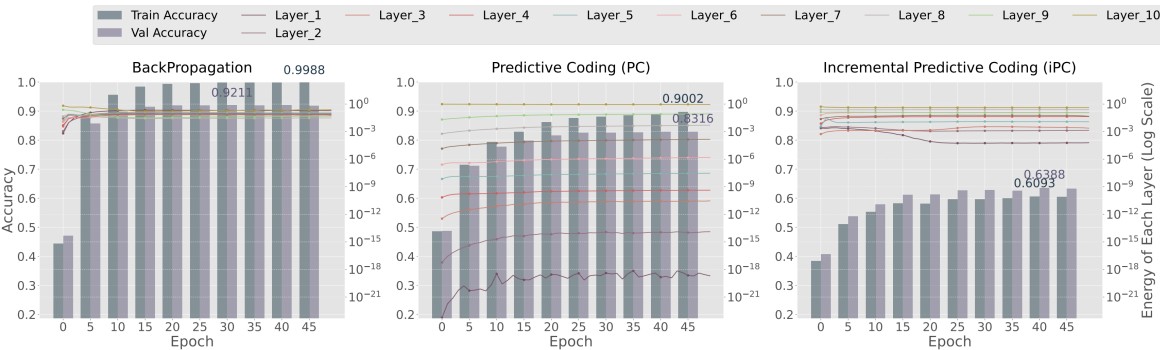

Figure 2. Normalized layer-wise energy distribution and accuracy comparison between BP and PCNs in a VGG10 on the CIFAR10 dataset. Colored curves represent the total energy of the individual layers of the model (or, the squared error of every layer for BP). The vertical lines represent the train and test accuracies of the model.

**Empirical Observation.** In models trained with PC, there is a significant energy imbalance, where early layers have up to 18 orders of magnitude less energy than later layers, while in models trained with iPC this phenomenon is less pronounced but still present, as the layer with less energy has an energy of about $10^{-5}$. This does not happen in BP-trained models, which exhibit a more uniform energy distribution across layers (above $10^{-2}$).

This set of experiments identifies the potential reason why deep PC models do not perform well. We formally link this phenomenon to the *spectral attenuation* of the Jacobian product in deep hierarchies (see theoretical proof in **Appendix C.1**). The following methods are designed specifically to mitigate this theoretically identified bottleneck.

### 4.2. Algorithmic Contributions

**Spiking Precision.** Training predictive coding models involves a critical trade-off between stability and the efficient propagation of error signals. On the one hand, large learning rates $\alpha$ for the neural activities can lead to instabilities: most of the best results in the field have been obtained using a small learning rate, such as $\alpha = 0.05$ (Pinchetti et al., 2025). On the other hand, such small learning rates can exponentially slow down the propagation of error signals across model layers, as noted in the supplementary material of (Song et al., 2020). To this end, we propose to modulate the precision having a *spike* proportional to the learning rate the first time the energy — initially concentrated in the output neurons — reaches a specific layer. In terms of temporal scheduling, this happens at time step $t = L - l$. For a network with $L$ layers and $T$ inference steps, the proposed spiking precision hence is:

$$\Sigma_t^l = \begin{cases} \alpha & \text{when } t = L - l, \\ 1, & \text{otherwise.} \end{cases} \quad (4)$$

Intuitively, the spikes allow the energy to be well propagated from the last to the first layer during the first $L$ iterations,

with the other updates happening as usual. Theoretically, this counteracts the spectral attenuation of the backward pass, ensuring that the effective update signal remains $\mathcal{O}(1)$ even in deep hierarchies (see proof in Appendix C.2).

**Forward Updates.** Due to large prediction errors that we find in the last layers, the neural activities observed at the end of the inference process tend to significantly deviate from their initial feed-forward values. But the feedforward values are the ones that are then used for predictions. We hence conjecture that synaptic weight updates based on $\mathbf{x}_T^l$ could potentially introduce errors that accumulate with network depth, leading to performance degradation in deeper architectures. To assess whether the proposed conjecture is correct, we introduce a new method for updating the weight parameters that uses both the starting and final states of neurons, according to the energy function defined as

$$\tilde{E}_T = \frac{1}{2} \sum_l \frac{\|\tilde{\varepsilon}_T^l\|^2}{\Sigma_t^l}, \qquad \text{where} \quad \tilde{\varepsilon}_T^l = \mathbf{x}_T^l - \mu_0^l. \quad (5)$$

Our method makes sure that weight adjustments stay connected to the initial feed-forward predictions while incorporating the refined representations obtained through iterative inference. In Appendix C.3, we demonstrate that Forward Update formulates a correction to the input drift problem, ensuring that updates are anchored to the feed-forward manifold.

#### 4.2.1. EXPERIMENTAL VALIDATION OF THE ALGORITHMIC CONTRIBUTIONS

Our hypothesis is that the proposed methods improve the performance of PC and iPC on deep models. To this end, we perform experiments on VGG-like models (Simonyan & Zisserman, 2015) — convolutional models followed by feedforward layers — on the CIFAR10 dataset, where we observe their test accuracies as a function of their depth. We again use backprop, PC, and iPC as baselines, and report the

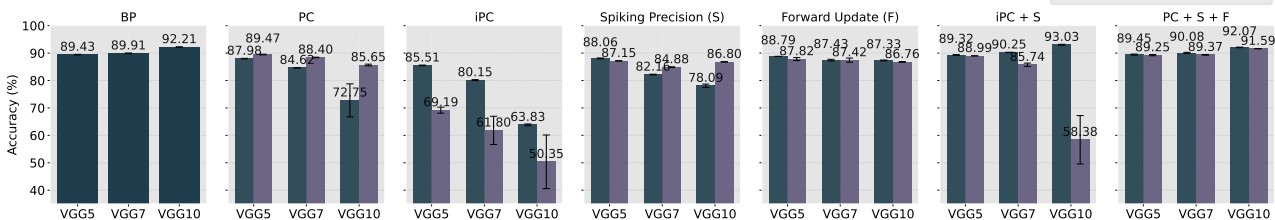

*Figure 3.* Test accuracies of various algorithms on the CIFAR10 dataset, evaluated on models of different depths. From the second plot onward, each pair of bars compares the performance of the algorithm with and without center nudging (CN).

results computed with and without centered nudging in Figure 3. All the details needed to reproduce the experiments, architectures, and hyperparameters used can be found in the Appendix D.

**Depth Scalability Results.** The barplots show that PC and iPC with and without center nudging significantly drop in test accuracy when the depth of the model is increased. In contrast, our proposed methods avoid the accuracy degradation as model depth increases. Particularly, iPC with spiking precision is the method that exhibits performance comparable to BP across VGG5/7/10 models, despite being completely local in space and time. We will later see that this is consistent with the more complex experiments, where this method is still the one achieving the best performance overall.

**On the role of Nudging.** In models without forward update, consistent with previous findings, using center nudging enhances algorithm accuracy, with more pronounced effects as the number of model layers increases in most cases. However, we found that once forward update is added, center nudging ceases to yield performance benefits. We provide a theoretical analysis of this phenomenon in Appendix C.3.2, showing that center nudging acts as a crude regularization to constrain the representation drift, a problem that is more fundamentally resolved by the gradient alignment property of forward updates.

**Analysis of Precision Schedules.** To address the problem of energy imbalances, we proposed *spiking precision* that help propagate the energy to the first layers. However, alternative options would have been to attenuate the energy accumulation in the last layers by gradually reducing their precision over time. We therefore evaluate two variants where precisions decay either linearly or exponentially with the number of time steps, using the same setup of the experiment above, that is, a VGG10 on CIFAR10.

The results, reported in Tab. 1, show that such decays provide improvements over the baseline, confirming the benefit of dynamically modulating precisions. However, spiking precisions consistently outperform both decay schedules,

underscoring that actively boosting error propagation (via spikes) is more effective than passively damping energy imbalances (via decay). We refer to the Appendix. F.2.1 for a more detailed description.

*Table 1.* Performance of different precision schedules on VGG10/CIFAR-10.

| Method | Test Accuracy (%) |
|---|---|
| Fixed | $87.33^{\pm 0.14}$ |
| Linear Decay | $88.35^{\pm 0.12}$ |
| Exponential Decay | $89.43^{\pm 0.18}$ |
| Spiking Precision | $92.07^{\pm 0.10}$ |

### 4.3. Architecture Adaptation

**Residual Connections.** Previously, we showed that our proposed methods overcome the depth limitation when training VGG-style models with 10 layers. However, this improvement does not extend to ResNet10, where performance still drops catastrophically. We conjecture that this degradation arises from a mismatch in how energy propagates through residual connections: the skip path delivers energy to higher layers earlier than the main feedforward path. Concretely, consider the residual block in Fig. 4(a): when the neurons $\mathbf{x}^{l+2}$ receive energy at time $t$, in the next step they propagate it both to $\mathbf{x}^{l+1}$ (through the feedforward connection) and to $\mathbf{x}^{l-1}$ (through the skip connection). As a result, higher-level neurons begin updating before the main stream of energy arrives.

Having two streams of energy that reach the same levels in different time steps contrasts with both the philosophy behind the spiking updates, that are supposed to boost the neural activities the first time this is reached by the energy, and iPC, that updates some weight parameters before the information that goes through the model has reached them. It has been shown that it makes the feedback signal diverge from the one of backprop (Salvatori et al., 2022). To address this temporal mismatch, we introduce auxiliary families of neurons along each residual connection, one for every skipped layer, as shown in Fig. 4(a). These auxiliary units act as buffers that delay the propagation of energy through

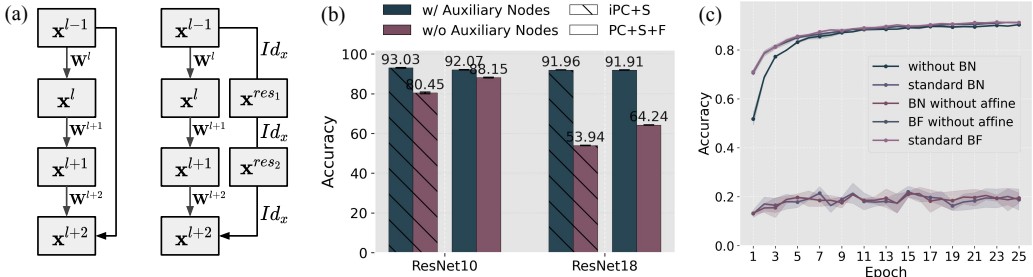

*Figure 4.* (a): A sketch of a residual block (left) and our proposed variation (right) with auxiliary neural activities, that prevent the error signal to travel from $\mathbf{x}^{l+2}$ to $\mathbf{x}^{l-1}$ in one timestep; (b) A barplot showing the gap in test accuracy on the CIFAR10 dataset between models with and without added neural activities on a ResNet18. Both plots refer to the best test accuracies reached by PC and iPC with the novel methods presented above. (c) Shows the test accuracy between models with the standard formulation of BN, without BN, and our proposed BF.

the skip path, ensuring that energy arriving via the residual connection reaches the start of the block synchronously with the energy propagated through the main path.

Formally, we augment the network with a total of $K$ auxiliary nodes $\mathbf{x}^{\mathbf{res_k}}$ distributed across the residual pathways. Here, $K$ is defined as the cumulative number of intermediate layers bypassed by all skip connections in the architecture. In this formulation, the prediction of every auxiliary node $\mu_t^{res_k}$ is defined by a fixed identity mapping from the preceding node, i.e., $\mu_t^{res_k} = Id(\mathbf{x_t^{res_{k-1}}})$. This structural constraint forces the feedback signal to traverse the residual path sequentially, thereby matching the propagation delay of the main pathway. The global energy $E_t$ is thus expanded to include the precision-weighted errors of these auxiliary variables:

$$E_t = \frac{1}{2} \sum_{l=1}^{L} \frac{\|\mathbf{x}_t^l - \mu_t^l\|^2}{\Sigma_t^l} + \frac{1}{2} \sum_{k=1}^{K} \frac{\|\mathbf{x}_t^{res_k} - \mu_t^{res_k}\|^2}{\Sigma_t^{res_k}}.$$

In the case of spiking updates, the covariance $\Sigma_t^{res_k}$ of an auxiliary node is set equal to the covariance $\Sigma_t^l$ of the main layer, ensuring unified energy propagation. This structural constraint forces the feedback signal to traverse the residual path sequentially, thereby matching the propagation delay of the main pathway and ensuring *phase synchronization* (see analysis in Appendix C.4).

**BatchNorm Freezing (BF).** BatchNorm has proven instrumental in stabilizing the training of deep neural networks, as it mitigates gradient-related issues and ensures smooth gradient propagation. During training, it achieves this by normalizing layer activations through the function

$$\text{BN}(x) = \gamma\left(\frac{x - \mu_B}{\sqrt{\sigma_B^2 + \epsilon}}\right) + \beta, \qquad (6)$$

where $\gamma$ and $\beta$ are learnable parameters, and $\mu_B$ and $\sigma_B^2$ are the mean and variance of the minibatch $B$. At test time, it uses statistics $\mu_r$ and $\sigma_r^2$ learned through exponential moving averages. However, when applied directly to

PCNs, BatchNorm fails to yield similar improvements. We hypothesize that this failure results from the iterative inference phase, where processing the same batch multiple times leads to a possible overfitting of the layer statistics. To address this issue, we propose BatchNorm Freezing (BF), a modification that freezes the states of the BatchNorm state during the inference phase, and updates running statistics exclusively during the learning phase while still using batch statistics for normalization during inference iterations.

### 4.3.1. VALIDATION OF THE ARCHITECTURE ADAPTATION

With the goal of validating the conjectures outlined above, we conducted two sets of experiments on the CIFAR-10 dataset. First, to assess the impact of *Auxiliary Neurons*, we trained a ResNet10 and a ResNet18 models both with and without these units, using our best-performing training schemes (iPC with spiking updates, and PC with spiking plus forward updates). Second, to evaluate *BatchNorm Freezing (BF)*, we revisited the VGG10 architecture and compared different forms of batch normalization, including our newly proposed BF variant. The results of both experiments are reported in Fig. 4(b) and (c), respectively.

**Results on Residual Connections.** For ResNet10, the results indicate that introducing auxiliary neurons resolves the skip-connection issue, enabling both algorithms to match the performance of the VGG10 baseline reported previously. This result is even starker in ResNet18, where the presence of multiple skip connections causes a much larger degradation of performance. Again, the addition of auxiliary neurons completely solves the problem, as the proposed models reach almost 92% test accuracy, compared to a maximum of 64%. This empirically confirms that phase synchronization (via auxiliary buffers) is a prerequisite for training deep residual PCNs.

*Table 2.* Test accuracies across datasets and architectures. We report mean and standard deviation over 5 runs.

| Algorithm | VGG5 | VGG7 | VGG10 | ResNet-10 | ResNet-18 |
|---|---|---|---|---|---|
| **CIFAR10** | | | | | |
| BP | $90.01^{\pm.15}$ | $91.32^{\pm.14}$ | $92.68^{\pm.10}$ | $92.93^{\pm.11}$ | $93.13^{\pm.16}$ |
| PC | $87.98^{\pm.11}$ | $84.62^{\pm.10}$ | $76.22^{\pm.43}$ | $69.85^{\pm2.1}$ | $15.63^{\pm7.2}$ |
| iPC | $86.01^{\pm.10}$ | $80.15^{\pm.18}$ | $63.83^{\pm.33}$ | $62.34^{\pm.27}$ | $21.90^{\pm1.5}$ |
| iPC+S | $89.73^{\pm.06}$ | $91.12^{\pm.08}$ | $93.03^{\pm.18}$ | $92.39^{\pm.04}$ | $91.96^{\pm.07}$ |
| PC+S+F | $89.45^{\pm.18}$ | $90.89^{\pm.04}$ | $93.27^{\pm.10}$ | $92.47^{\pm.01}$ | $91.93^{\pm.14}$ |
| **CIFAR100 (Top-1)** | | | | | |
| BP | $67.39^{\pm.25}$ | $67.67^{\pm.11}$ | $71.25^{\pm.21}$ | $71.21^{\pm.09}$ | $71.69^{\pm.21}$ |
| PC | $61.84^{\pm.18}$ | $56.80^{\pm.14}$ | $50.76^{\pm.37}$ | $41.51^{\pm.32}$ | $1.59^{\pm.02}$ |
| iPC | $56.07^{\pm.16}$ | $43.99^{\pm.30}$ | $31.99^{\pm.17}$ | $22.91^{\pm.23}$ | $1.56^{\pm.24}$ |
| iPC+S | $66.91^{\pm.12}$ | $67.10^{\pm.12}$ | $69.84^{\pm.17}$ | $70.02^{\pm.24}$ | $70.38^{\pm.20}$ |
| PC+S+F | $67.16^{\pm.16}$ | $67.71^{\pm.10}$ | $72.02^{\pm.12}$ | $71.30^{\pm.21}$ | $70.90^{\pm.18}$ |
| **TinyImageNet (Top-1)** | | | | | |
| BP | $47.81^{\pm.12}$ | $50.13^{\pm.06}$ | $53.61^{\pm.12}$ | $53.02^{\pm.20}$ | $58.18^{\pm.12}$ |
| PC | $41.29^{\pm.20}$ | $41.15^{\pm.14}$ | $31.87^{\pm.03}$ | $13.59^{\pm.12}$ | $0.84^{\pm.02}$ |
| iPC | $29.94^{\pm.47}$ | $19.76^{\pm.15}$ | $11.41^{\pm.23}$ | $11.66^{\pm.39}$ | $1.44^{\pm.05}$ |
| iPC+S | $48.13^{\pm.10}$ | $48.75^{\pm.12}$ | $52.40^{\pm.20}$ | $54.94^{\pm.30}$ | $57.83^{\pm.21}$ |
| PC+S+F | $49.35^{\pm.09}$ | $50.64^{\pm.12}$ | $55.31^{\pm.25}$ | $53.25^{\pm.27}$ | $54.24^{\pm.63}$ |
| **TinyImageNet (Top-5)** | | | | | |
| BP | $72.15^{\pm.12}$ | $73.94^{\pm.10}$ | $77.11^{\pm.10}$ | $72.90^{\pm.16}$ | $79.94^{\pm.06}$ |
| PC | $66.68^{\pm.09}$ | $66.25^{\pm.11}$ | $58.14^{\pm.04}$ | $37.99^{\pm.08}$ | $5.34^{\pm.01}$ |
| iPC | $54.73^{\pm.52}$ | $40.36^{\pm.22}$ | $30.42^{\pm.36}$ | $26.51^{\pm.71}$ | $11.58^{\pm.21}$ |
| iPC+S | $72.71^{\pm.12}$ | $73.39^{\pm.10}$ | $76.76^{\pm.15}$ | $78.88^{\pm.32}$ | $77.55^{\pm.12}$ |
| PC+S+F | $73.46^{\pm.09}$ | $75.63^{\pm.08}$ | $79.30^{\pm.17}$ | $77.72^{\pm.23}$ | $74.70^{\pm.47}$ |

**Results on Normalization.** In the experiments with BF, we observe that standard Batch Normalization fails to support convergence in PCNs. In contrast, our BF variant not only maintains convergence but also improves accuracy. This indicate that stabilizing activation distributions plays a crucial role in supporting energy-based optimization within predictive coding architectures.

In the next section, we will demonstrate that these combined algorithmic and structural improvements allow PCNs to scale to even deeper models and more complex datasets.

# 5. Larger Scale Experiments

In this section, we demonstrate that our proposed framework can reach performance comparable to that of BP across complex datasets and deep architectures. To provide a comprehensive evaluation, we test them on CIFAR-10/100 (Krizhevsky et al., 2009) and TinyImageNet (Le & Yang, 2015) using VGG-like models and ResNet architectures (He et al., 2016). For all experiments, we performed an extensive hyperparameter search and report the best test accuracy averaged over 5 runs. Detailed architectures, reproducibility details are provided in Appendix D.

**Main Results on Depth.** We report a comprehensive comparison in Table 2. As expected, while in shallow models all methods can either match or approximate the performance

of BP, baseline approaches degrade significantly as depth increases. In contrast, our newly proposed methods maintain robust performance. However, a closer inspection reveals that 'iPC+S' consistently underperforms 'PC+S+F', particularly in deeper architectures. To investigate the source of this disparity, we provide an ablation study in *Appendix F* which reveals that while BF is essential for stabilizing standard PC, it yields no benefit for iPC, and even leads to slight degradation in some deep models. We provide a theoretical analysis of this interference between continuous updates and frozen statistics in *Appendix F.5.1*.

**Full Factorial Ablation.** To enable clear attribution of gains to individual components, we conducted a full factorial ablation across all proposed methods: Spiking Precision (S), Forward Update (F), and BatchNorm Freezing (BF). The results on CIFAR10 and CIFAR100 are reported in Table. 8, and a radar chart visualizing relative component contributions is provided in Figure. 5. The key findings are: (1) Spiking Precision and Forward Update matter most for deep sequential models, and their contribution grows with depth; (2) Spiking Precision is crucial for residual networks, as skip connections create a learning rate conflict that it resolves by dynamically modulating the effective gain; (3) BatchNorm Freezing is essential for standard PC to prevent overfitting of running statistics during iterative relaxation; (4) Auxiliary Neurons resolve the temporal phase mismatch inherent in skip connections. On more complex datasets (CIFAR100 vs. CIFAR10), the combination of all components becomes increasingly necessary. A more detailed per-component analysis with energy propagation plots is provided in Appendix F.

Beyond these component analyses, we further includes a detailed *hyperparameter analysis in Appendix I* and a stress-test on an extremely deep *101-layer ResMLP in Appendix G*).

*Table 3.* Test accuracies (%) on Galaxy10 DECals and ImageNet32 using ResNet-18 and VGG-15, respectively.

| Algorithm | Galaxy10 | ImgNet32 (Top-1) | ImgNet32 (Top-5) |
|---|---|---|---|
| BP | $84.60^{\pm0.10}$ | 37.21 | 61.56 |
| PC+S+F | $84.37^{\pm0.10}$ | 36.09 | 60.92 |
| iPC+S | $84.08^{\pm0.12}$ | 36.23 | 61.52 |

**Scaling to Resolution and Complexity.** To evaluate the scalability of our approach to higher-dimensional inputs and more complex data, we conducted experiments on the Galaxy10 DECals dataset (Dey et al., 2019; Lintott et al., 2008) and ImageNet32 (Chrabaszcz et al., 2017). Galaxy10 tests robustness to increased input resolution, while ImageNet32 preserves the semantic complexity of ImageNet-1k with 1,000 classes at a reduced resolution ($32 \times 32$). As shown in Table 3, our methods (PC+S+F and iPC+S)

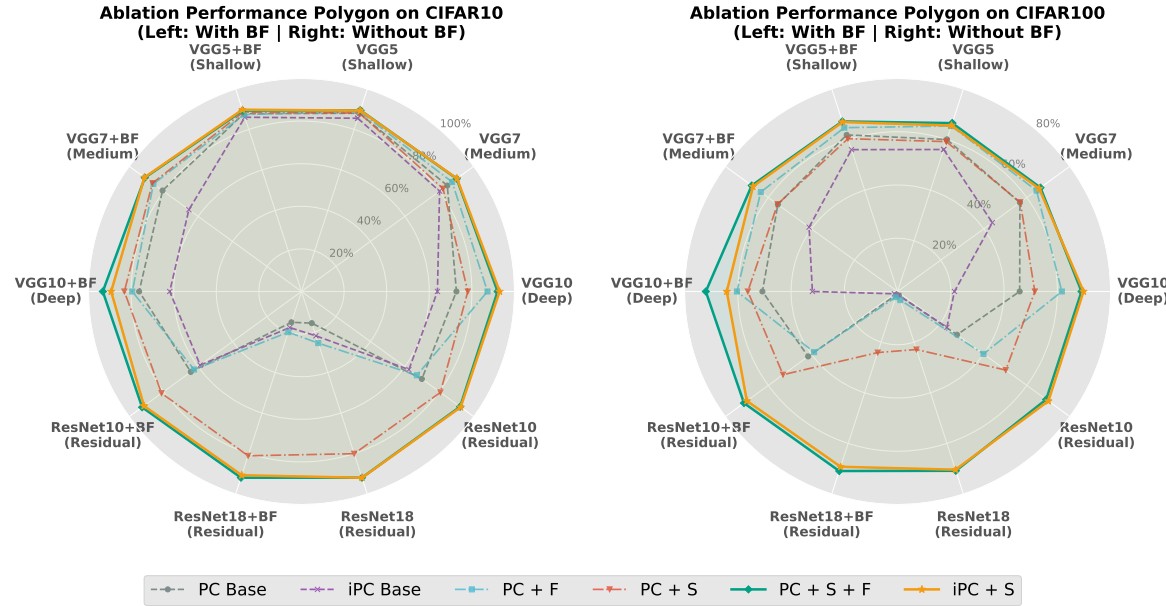

*Figure 5.* Ablation performance radar chart comparing the relative contributions of different components (S, F, BF) across sequential (VGG) and residual (ResNet) architectures on CIFAR10 and CIFAR100.

achieve performance close to the backpropagation baseline on Galaxy10 and competitive Top-1 and Top-5 accuracies on ImageNet32, demonstrating that predictive coding networks scale effectively to both higher-dimensional inputs and large-scale, multi-class distributions.

## 6. Conclusion

In this work, we have investigated the following research question: Why do deep models trained with the predictive coding energy fail to match the accuracy of their counterparts trained with backpropagation? We have addressed this problem from both the algorithmic and the architectural sides. Algorithmically, we have proposed both a novel technique that leverages a dynamical precision-weighting of prediction errors to better regularize the energy landscape, and a novel weight update mechanism. From the architecture adaptation side, we have shown how to modify the residual connections to allow the training of PC-based ResNets, and developed a more effective normalization technique. The results show that we are now able to train VGG models with 15 layers, and ResNet18 on Tiny Imagenet.

A limitation of our approach is that forward updates require access to prediction values at initialization, introducing a degree of biological implausibility due to non-locality in time. Future work will explore more biologically plausible weight updates to address this issue, with recent theoretical results on training very deep feedforward models providing a promising direction (Innocenti et al., 2026). Despite this limitation, iPC with spiking updates achieves comparable performance, demonstrating that complex models such as ResNet-18 can be trained using learning rules that are local in time and space while matching backpropagation performance.

## Acknowledgments

This research was funded in part by the AXA Research Fund and in part by the Austrian Science Fund (FWF) 10.55776/COE12.

## Impact Statement

This work is a foundational study that bridges the gap between neuroscience and artificial intelligence by enabling Predictive Coding Networks to train deep architectures effectively. By resolving fundamental stability bottlenecks in Predictive Coding, our research fosters advancements in energy-efficient, brain-inspired computing. Besides that, there are no major societal impact to state.

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

## A. Appendix

Here we provide an explanation of the performed experiments, as well as a detailed description of all the parameters needed to replicate the results of the paper. We also provide an ablation study that shows the performance of the individual methods in isolation.

## B. Algorithm

The pseudocode is provided in Algorithm 1.

The proposed training procedure is detailed in Algorithm 1. This algorithm presents a unified framework for training both Predictive Coding (PC) and Incremental Predictive Coding (iPC) models, incorporating proposed techniques such as Precision Schedules and Forward Update. The entire process is structured into three core phases:

*Initialization:* The network's neuronal activities are initialized through a single feed-forward pass.

*Inference Learning (Relaxation Phase):* With the input and target values clamped, the hidden layer activities $(x_t^l)$ iteratively relax over T steps to minimize prediction errors across the hierarchy. For iPC, the weights$(\mathbf{W}_t^l)$ and neural activities $(x_t^l)$ are jointly updated within each iteration of this relaxation phase.

*Weight Update:* After the inference phase concludes, the PC model computes and applies its weight updates based on the final states and the initial predictions, following the Forward Update rule.

## C. Theoretical Analysis

### C.1. Theorem 1: The Iterative Signal Decay

**Problem Statement:** In the inference phase (Phase 2 of Algorithm 1), error signals must propagate from the clamped output layer $\mathbf{x}^L = \mathbf{y}$ down to the shallow layers $l$ to update $\mathbf{x}^l$. We analyze the magnitude of the effective driving signal received by layer $l$ as a function of the inference learning rate $\alpha$ and the network depth.

**Theorem C.1** (Iterative Signal Attenuation). *Consider the update rule for hidden layer activity $\mathbf{x}^l$ at inference step $t$:*

$$\mathbf{x}_{t+1}^l \leftarrow \mathbf{x}_t^l + \frac{\alpha}{\Sigma_{t+1}^l}\varepsilon_{t+1}^l - \frac{\alpha}{\Sigma_{t+1}^l}(\mathbf{W}^{l+1})^\top(\varepsilon_{t+1}^{l+1} \odot f'(\mathbf{x}_t^l)). \tag{7}$$

*We define the "top-down drive" $\mathcal{D}^{l+1\rightarrow l}$ as the component of the update derived from the layer above. Under standard assumptions (bounded weights and activation derivatives), the magnitude of the signal originating from the output layer $L$ and arriving at layer $l$ after the necessary number of time steps decays geometrically:*

$$\|Signal^{L\rightarrow l}\| \propto \alpha^{L-l}\|\varepsilon^L\|. \tag{8}$$

*For standard hyperparameters where $\alpha \ll 1$ (e.g., $\alpha = 0.1$) and fixed precision $\Sigma = 1$, this results in exponential signal decay for deep networks.*

*Proof.* We trace the propagation of the supervised error signal $\varepsilon^L$ (at the output) to the latent activities of lower layers through the iterative process defined in Algorithm 1.

**Step 1: Signal Injection at Output.** At the output layer $L$, the activity is clamped to the target $\mathbf{y}$. The prediction error at the output is $\varepsilon^L = \mathbf{y} - \boldsymbol{\mu}^L$. This error term is non-zero and serves as the source of the global learning signal.

**Step 2: Single-Step Transmission.** Consider the update for layer $L - 1$. The term that introduces information from the layer above is:

$$\Delta\mathbf{x}_{top-down}^{L-1} = -\frac{\alpha}{\Sigma^{L-1}}(\mathbf{W}^L)^\top(\varepsilon^L \odot f'(\mathbf{x}^{L-1})). \tag{9}$$

Assuming fixed precision $\Sigma = 1$, the magnitude of this update is scaled by $\alpha$. Specifically, the "influence" of $\varepsilon^L$ on $\mathbf{x}^{L-1}$ is proportional to $\alpha\|\mathbf{W}^L\|$.

---

**Algorithm 1** Training a PC/iPC with Precision Schedules and Forward Update

---

**Require:** Input data $\mathbf{o}$, target $\mathbf{y}$, weights $\{\mathbf{W}^l\}_{l=1}^L$, activation function $f(\cdot)$, learning rates $\eta, \alpha$, relaxation steps $T$, precision schedule $\Sigma_t^l$

1: **// Phase 1: Initialization**
2: $\mathbf{x}_0^0 \leftarrow \mathbf{o}$
3: **for** $l = 1$ **to** $L$ **do**
4:     {*Initial feed-forward pass*}
5:     $\boldsymbol{\mu}_0^l \leftarrow \mathbf{W}^l f(\mathbf{x}_0^{l-1})$
6:     $\mathbf{x}_0^l \leftarrow \boldsymbol{\mu}_0^l$
7: **end for**
8: **// Phase 2: Inference Learning (Relaxation)**
9: Clamp input: $\mathbf{x}_t^0 \leftarrow \mathbf{o}$ for $t \in [1, T]$
10: Clamp output: $\mathbf{x}_t^L \leftarrow \mathbf{y}$ for $t \in [1, T]$
11: **for** $t = 0$ **to** $T - 1$ **do**
12:     **for** $l = 1$ **to** $L - 1$ **do**
13:         {*Update hidden layer activities*}
14:         $\boldsymbol{\mu}_{t+1}^l \leftarrow \mathbf{W}^l f(\mathbf{x}_{t+1}^{l-1})$
15:         $\boldsymbol{\varepsilon}_{t+1}^l \leftarrow \mathbf{x}_t^l - \boldsymbol{\mu}_{t+1}^l$
16:         $\boldsymbol{\varepsilon}_{t+1}^{l+1} \leftarrow \mathbf{x}_t^{l+1} - \mathbf{W}^{l+1} f(\mathbf{x}_t^l)$
17:         $\Delta \mathbf{x}_t^l \leftarrow \frac{\alpha}{\Sigma_{t+1}^l} \left( \boldsymbol{\varepsilon}_{t+1}^l - (\mathbf{W}^{l+1})^\top \boldsymbol{\varepsilon}_{t+1}^{l+1} \odot f'(\mathbf{x}_t^l) \right)$
18:         $\mathbf{x}_{t+1}^l \leftarrow \mathbf{x}_t^l + \Delta \mathbf{x}_t^l$
19:         **if** iPC **then**
20:             {*Update weights if in iPC Training*}
21:             $\Delta \mathbf{W}_{t+1}^l \leftarrow \frac{\eta}{\Sigma_{t+1}^l} \left( \boldsymbol{\varepsilon}_{t+1}^l f(\mathbf{x}_{t+1}^{l-1})^\top \right)$
22:             $\mathbf{W}^l \leftarrow \mathbf{W}^l - \Delta \mathbf{W}_{t+1}^l$
23:         **end if**
24:     **end for**
25: **end for**
26: **// Phase 3: Learning (Weight Update for PC)**
27: **if** PC **then**
28:     **for** $l = 1$ **to** $L$ **do**
29:         $\tilde{\boldsymbol{\varepsilon}}_T^l \leftarrow \mathbf{x}_T^l - \boldsymbol{\mu}_0^l$ {*Calculate Forward Update error*}
30:         $\Delta \mathbf{W}^l \leftarrow \eta \cdot \tilde{\boldsymbol{\varepsilon}}_T^l f(\mathbf{x}_0^{l-1})^\top$
31:         $\mathbf{W}^l \leftarrow \mathbf{W}^l - \Delta \mathbf{W}^l$
32:     **end for**
33: **end if**

---

**Step 3: Recursive Propagation.** For this signal to influence layer $L - 2$, $\mathbf{x}^{L-1}$ must first change, which then updates $\varepsilon^{L-1}$, which finally drives $\mathbf{x}^{L-2}$. Let $\mathcal{T}_{k+1 \to k}$ be the transmission operator from layer $k + 1$ to $k$. From the update rule, the gain of this operator is roughly $\alpha \|(\mathbf{W}^{k+1})^\top\|$. For the signal to travel from $L$ to $l$, it must pass through $L - l$ such transmission steps sequentially. The total effective gain is the product of layer-wise gains:

$$\text{Gain}(L \to l) \approx \prod_{k=l}^{L-1} \left( \alpha \|(\mathbf{W}^{k+1})^\top\| \cdot \|f'(\mathbf{x}^k)\| \right). \tag{10}$$

Let $\gamma = \mathbb{E}[\|\mathbf{W}\| \cdot \|f'\|]$ be the spectral norm of the effective weight matrix. The signal strength at layer $l$ is:

$$S_l \approx (\alpha \cdot \gamma)^{L-l} \|\boldsymbol{\varepsilon}^L\|. \tag{11}$$

**Mechanism of Failure.** Standard PC requires a small learning rate $\alpha$ to ensure the stability of the discretization of the differential equation (typically $\alpha \in [0.01, 0.1]$). Assuming $\gamma \approx 1$ (standard initialization):

- For $l = L - 1$ (1 layer down): Decay is $\alpha^1$.

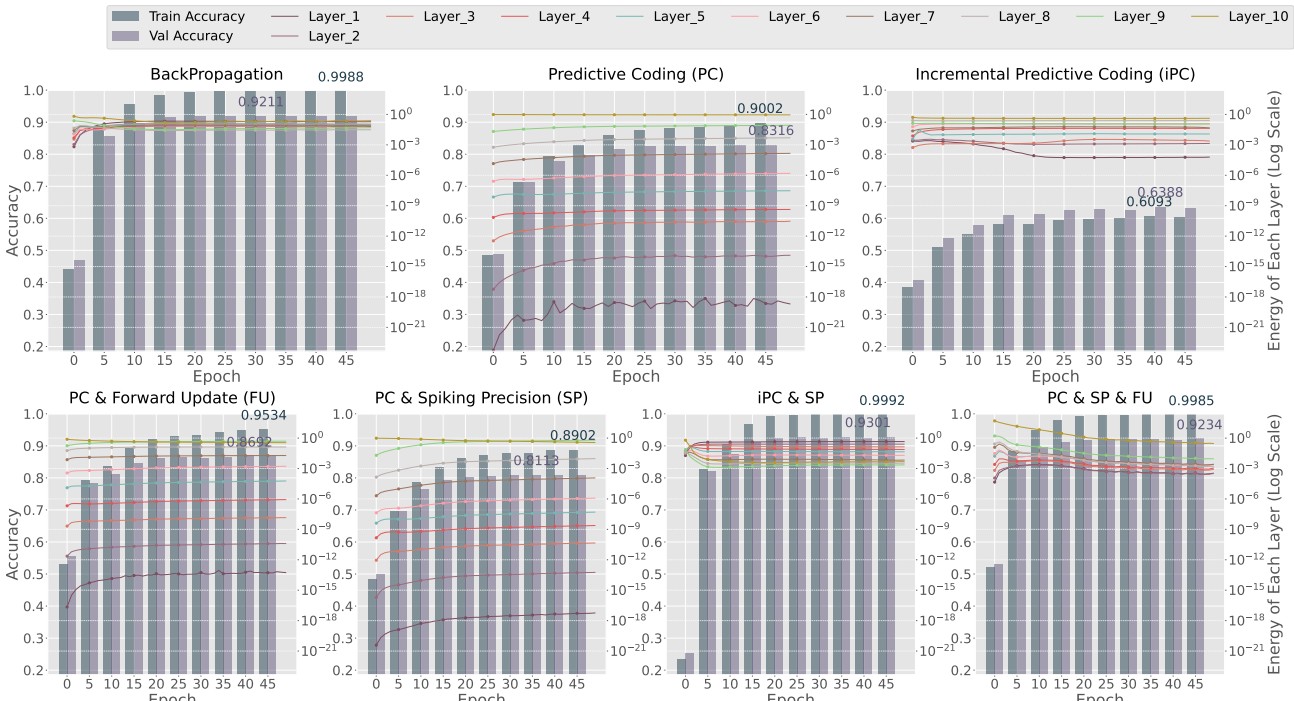

*Figure 6.* Normalized layer-wise energy distribution and accuracy comparison between BP and PCNs in a VGG10 on the CIFAR10 dataset. Colored curves represent the total energy of the individual layers of the model (or, the squared error of every layer for BP). The vertical lines represent the train and test accuracies of the model.

- For $l = L - 10$ (10 layers down): Decay is $\alpha^{10}$.

Substituting $\alpha = 0.05$ yields a decay factor of $\alpha^{10} \approx 9.7 \times 10^{-14}$. This calculation provides a formal explanation for the empirical energy imbalance: the driving signal from the label is exponentially attenuated by the repeated multiplication of the small learning rate $\alpha$ before reaching the early layers, a theoretical result that is consistent with the observations in Figure 6.

### C.2. Mechanism 1: Spiking Precision as Gain Restoration

To counteract the iterative attenuation caused by the small inference learning rate $\alpha$ (typically $\alpha \approx 0.05$), standard precision schedules are insufficient. We introduce *Spiking Precision*, which dynamically modulates the effective step size.

**Theorem C.2** (Gain Restoration). *Let the spiking precision schedule define an effective feedback gain $\lambda = \frac{\alpha}{\Sigma_{spike}}$ at the moment of signal arrival ($t = L - l$). We treat $\lambda$ as a hyperparameter and perform a search over a broad logarithmic spectrum (e.g., $\lambda \in (5 \times 10^{-3}, 9 \times 10^{-1})$). We empirically prove that the optimal performance consistently emerges in the regime $\lambda \in (0.5, 0.9)$. This result confirms that restoring the feedback gain to unity order ($\mathcal{O}(1)$) is a necessary condition for scaling PCNs, effectively shifting the dynamics from an **exponential decay regime** to a **stable propagation regime**.*

*Proof.* Recall the update rule for the top-down signal transmission:

$$\Delta \mathbf{x}^l_{top-down} = - \underbrace{\frac{\alpha}{\Sigma^l_{t+1}}}_{\text{Effective Gain } \lambda} (\mathbf{W}^{l+1})^\top (\varepsilon^{l+1}_{t+1} \odot f'(\mathbf{x}^l_t)). \tag{12}$$

**Standard Regime (The Vanishing Trap).** In standard PC, $\Sigma = 1$, implying an effective gain $\lambda_{std} = \alpha$. As derived in Theorem 1, the signal passing through $D$ layers scales by $\lambda^D_{std} = \alpha^D$. For a deep network ($D = 10$) with $\alpha = 0.05$:

$$\text{Signal}_{std} \propto (0.05)^{10} \approx 10^{-14}. \tag{13}$$

The signal is numerically annihilated, leading to the observed energy imbalance.

**Spiking Regime (The Restoration).** With Spiking Precision, we decouple the gain from the learning rate $\alpha$. We search for an optimal $\lambda$ in a wide range $(0.005, 0.9)$. Our experiments reveal that the optimal $\lambda^*$ consistently falls in $(0.5, 0.9)$. Comparing the signal magnitude for an optimal $\lambda^* \approx 0.6$ versus the baseline $\alpha = 0.05$:

$$\frac{\text{Signal}_{spike}}{\text{Signal}_{std}} \approx \frac{(0.6)^{10}}{(0.05)^{10}} = \frac{0.006}{10^{-14}} \approx 6 \times 10^{11}. \tag{14}$$

Even though $\lambda^*$ is not strictly 1, it is sufficiently large to boost the bottom-layer energy by *11 orders of magnitude*. The fact that the hyperparameter search naturally converges to the high-gain regime $(> 0.5)$ provides strong empirical evidence that the network intrinsically requires signal restoration to counteract the attenuation caused by $\alpha$, validating our theoretical analysis.

### C.3. Mechanism 2: Forward Update as Amortized Inference Anchoring

In deep Predictive Coding, we identify a critical misalignment between the *inference objective* (minimizing energy at step $T$) and the *learning objective* (improving predictions for the next forward pass). We demonstrate that standard PC updates optimize for a transient state that vanishes after reset, whereas our Forward Update (F) anchors the learning to the stable feed-forward manifold. We further show that this formulation renders heuristic constraints like Center Nudging (CN) redundant. Let $\mathbf{x}_0^l$ denote the initial feed-forward activity (the "cause") and $\mathbf{x}_T^l$ denote the relaxed activity after inference (the "target").

- **The Target ($\mathbf{x}_T$):** The relaxed state $\mathbf{x}_T$ is a "better" representation than $\mathbf{x}_0$ because it integrates both bottom-up sensory information and top-down supervision from the loss function. It represents *where the model should be*.

- **The Input Context:** Crucially, during the testing phase or the beginning of the next training iteration, the network is reset. The weight $\mathbf{W}^l$ will receive $\mathbf{x}_0^{l-1}$ as input, not $\mathbf{x}_T^{l-1}$.

**Standard PC Misalignment.** Standard PC updates weights to minimize the prediction error at equilibrium:

$$\mathcal{L}_{PC} = \frac{1}{2}\|\mathbf{x}_T^l - \mathbf{W}^l f(\mathbf{x}_T^{l-1})\|^2 \implies \Delta\mathbf{W}_{PC} \propto (\mathbf{x}_T^l - \boldsymbol{\mu}_T^l) f(\mathbf{x}_T^{l-1})^\top. \tag{15}$$

This rule optimizes the weights to map the *drifted input* $\mathbf{x}_T^{l-1}$ to the target $\mathbf{x}_T^l$. However, since $\mathbf{x}_T^{l-1}$ is a temporary state that disappears once inference ends, minimizing this error does not guarantee that the prediction will improve when the network is reset to $\mathbf{x}_0^{l-1}$.

#### C.3.1. FORWARD UPDATE (F): BRIDGING THE GAP

We propose that the goal of learning in PC should be **Amortized Inference**: training the generative weights to produce the optimized inference result $\mathbf{x}_T$ directly from the initial feed-forward state $\mathbf{x}_0$. Our proposed Forward Update minimizes the discrepancy between the inference target and the feed-forward prediction:

$$\mathcal{L}_F = \frac{1}{2}\|\mathbf{x}_T^l - \mathbf{W}^l f(\mathbf{x}_0^{l-1})\|^2 \implies \Delta\mathbf{W}_F \propto (\mathbf{x}_T^l - \boldsymbol{\mu}_0^l) f(\mathbf{x}_0^{l-1})^\top. \tag{16}$$

By anchoring the update to $\mathbf{x}_0$, we force the weights to encode the transition from the *actual input* the layer will see ($\mathbf{x}_0$) to the *improved representation* found by inference ($\mathbf{x}_T$). This ensures that the valuable global information captured in $\mathbf{x}_T$ is effectively "distilled" into the feed-forward weights, allowing the model to generate better predictions in a single pass.

#### C.3.2. REDUNDANCY OF CENTER NUDGING (CN)

This perspective explains why Center Nudging (CN) becomes obsolete under the Forward Update (F) rule. In standard PC, if the input $\mathbf{x}_T^{l-1}$ drifts too far from $\mathbf{x}_0^{l-1}$, the weight update becomes uncorrelated with the feed-forward task. Center Nudging acts as a heuristic regularizer by clamping the output (and indirectly hidden layers) closer to the initialization. It artificially reduces the distance $\|\mathbf{x}_T - \mathbf{x}_0\|$ to keep the standard PC update valid.

**F Solves the Root Cause.** The F rule explicitly handles the drift by defining the update vector as the difference between the target $\mathbf{x}_T$ and the anchor $\mathbf{x}_0$. It accepts $\mathbf{x}_T$ as the ideal target regardless of how far it has moved, but correctly attributes the cause to $\mathbf{x}_0$. Since F mathematically accounts for the disconnect between $\mathbf{x}_T$ and $\mathbf{x}_0$, the artificial constraints of CN are no longer required. In fact, using CN alongside F is detrimental because CN restricts $\mathbf{x}_T$ from fully absorbing the top-down supervision, limiting the "better" target that F tries to learn.

## C.4. Mechanism 3: Phase Synchronization in Residual Networks

In ResNets, the feedback signal traverses two paths with distinct latencies. We model this as a discrete-time dynamical system.

**Proposition C.3** (Phase Mismatch). *Let $\mathcal{Z}^{-1}$ be the unit delay operator such that $\mathcal{Z}^{-1}\mathbf{x}_t = \mathbf{x}_{t-1}$. In a standard PC residual block, the feedback signal $F_t^{l-1}$ received by layer $l-1$ is:*

$$F_t^{l-1} = \underbrace{\varepsilon_{t-1}^{l+1}}_{\text{Skip Path (Latency 1)}} + \underbrace{(\mathbf{W}^l)^\top \varepsilon_{t-1}^l}_{\text{Main Path}}. \tag{17}$$

*Since $\varepsilon^l$ itself depends on $\varepsilon_{t-1}^{l+1}$, the main path has an effective latency of 2 steps. The total signal is a superposition of asynchronous states:*

$$F_{total} \sim \mathcal{Z}^{-1}S_{skip} + \mathcal{Z}^{-2}S_{main}. \tag{18}$$

*This phase lag $\Delta\tau = 1$ induces high-frequency oscillations in the energy landscape, preventing convergence.*

**Solution (Auxiliary Nodes).** By introducing auxiliary nodes $\mathbf{x}_{res}$, we introduce an additional computational step in the skip connection. The skip path operator becomes $\mathcal{Z}^{-1} \circ \mathcal{Z}^{-1} = \mathcal{Z}^{-2}$. Now, $F_{total} \sim \mathcal{Z}^{-2}(S_{skip} + S_{main})$. The terms are now *phase-aligned* (temporally coherent), satisfying the condition for constructive interference of error signals. This provides a theoretical guarantee for the stability improvements observed in Figure. 4(b).

# D. Experiments Setting

**Model.** We conducted experiments on four VGG-based models: VGG5, VGG7, VGG10, and VGG15 and two ResNet-based models: ResNet-10, ResNet-18. The detailed architectures of these models are presented in Table 4.

*Table 4.* Detailed architectures of base models.

| VGG5 | VGG7 |
|---|---|
| Channel Sizes: [128, 256, 512, 512] | Channel Sizes: [128, 128, 256, 256, 512, 512] |
| Kernel Sizes: [3, 3, 3, 3] | Kernel Sizes: [3, 3, 3, 3, 3, 3] |
| Strides: [1, 1, 1, 1] | Strides: [1, 1, 1, 1, 1, 1] |
| Paddings: [1, 1, 1, 0] | Paddings: [1, 1, 1, 0, 1, 0] |
| Pool window: $2 \times 2$ | Pool window: $2 \times 2$ |
| Pool stride: 2 | Pool stride: 2 |
| Linear Layers: 1 | Linear Layers: 1 |
| **VGG10** | **VGG15** |
| Channel Sizes: [64, [128]x3, [256]x4, 512] | Channel Sizes: [64, 64, 128, 128, [256]x3, [512]x6] |
| Kernel Sizes: [3, 3, 3, 3, 3, 3, 3, 3, 3] | Kernel Sizes: [3, 3, 3, 3, 3, 3, 3, 3, 3, 3, 3, 3] |
| Strides: [1, 1, 1, 1, 1, 1, 1, 1, 1] | Strides: [1, 1, 1, 1, 1, 1, 1, 1, 1, 1, 1, 1, 1] |
| Paddings: [1, 1, 1, 1, 1, 1, 1, 1, 1] | Paddings: [1, 1, 1, 1, 1, 1, 1, 1, 1, 1, 1, 1, 1] |
| Pool window: $2 \times 2$ | Pool window: $2 \times 2$ |
| Pool stride: 2 | Pool stride: 2 |
| Linear Layers: 1 | Linear Layers: 2 |
| **ResNet10** | **ResNet18** |
| Initial Conv: 3x3, Stride 1, Channel Size 64 | Initial Conv: 3x3, Stride 1, Channel Size 64 |
| Res-Block: [1, 1, 1, 1] | Res-Block: [2, 2, 2, 2] |
| Channel Sizes: [64, 128, 256, 512] | Channel Sizes: [64, 128, 256, 512] |
| Strides: [1, 2, 2, 2] | Strides: [1, 2, 2, 2] |
| Linear Layers: 1 | Linear Layers: 1 |

**Experiments.** The benchmark results of above models are obtained with CIFAR10, CIFAR100 and Tiny ImageNet. The datasets are normalized as in Table 5.

*Table 5.* Data normalization.

|  | **Mean ($\mu$)** | **Std ($\sigma$)** |
|---|---|---|
| CIFAR10 | [0.4914, 0.4822, 0.4465] | [0.2023, 0.1994, 0.2010] |
| CIFAR100 | [0.5071, 0.4867, 0.4408] | [0.2675, 0.2565, 0.2761] |
| Tiny ImageNet | [0.485, 0.456, 0.406] | [0.229, 0.224, 0.225] |

For data augmentation on CIFAR10, CIFAR100, and Tiny ImageNet training sets, we use 50% random horizontal flipping. We also apply random cropping with different setups. For CIFAR10 and CIFAR100, images are randomly cropped to 32×32 resolution with 4-pixel padding. For Tiny ImageNet, images are randomly cropped to 64×64 resolution images with 8-pixel padding. For testing on those datasets, we applied only standard data normalization, without using any additional data augmentation techniques.

*Table 6.* Hyperparameters search configuration.

| **Parameter** | **PCNs** | **BP** |
|---|---|---|
| Epoch | 25 | |
| Batch Size | 128 | |
| Activation | [leaky relu, gelu, hard tanh, relu] | |
| $\alpha$ | [0.01, 0.05, 0.001, 0.005] | - |
| $\beta$ | [0.0, 1.0], 0.15[1] | - |
| $lr_x$ | (5e-3, 9e-1)[2] | - |
| $lr_w$ | (1e-5, 3e-2)[2] | (1e-5, 3e-4)[2] |
| $momentum_x$ | [0.0, 1.0], 0.1[1] | - |
| $weight\_decay_w$ | (1e-5, 1e-2)[2] | |
| T (VGG-5) | [5,6,7,8] | - |
| T (VGG-7) | [7,9,11,13] | - |
| T (VGG-10) | [10,12,14,16] | - |
| T (VGG-15) | [15,17,19,21] | - |
| T (ResNet-10) | [10,12,14,16] | - |
| T (ResNet-18) | [18,20,22,24] | - |

[1]: "[a, b], c" denotes a sequence of values from a to b with a step size of c. [2]: "(a, b)" represents a log-uniform distribution between a and b.

For the optimizer and scheduler, we employ mini-batch stochastic gradient descent (SGD) with momentum for updating $x$ during the relaxation phase. For the learning phase, we optimize weights $W$ using AdamW with weight decay. The learning rate schedule follows a warmup-cosine annealing pattern without restarts. This scheduler initiates training with a low learning rate during the warmup period, then smoothly transitions to a cosine-shaped decay curve, preventing abrupt performance degradation. The schedule parameters are configured as follows: the peak learning rate reaches 1.1 times the initial rate, the final learning rate settles at 0.1 times the initial rate, and the warmup phase spans 10% of the total iteration steps.

We conduct a rigorous hyperparameter search for all models, including the baselines, based on the search space specified in Table 6. All experiments were implemented using the PCX library, a JAX-based framework specifically designed for predictive coding networks that provides comprehensive benchmarking capabilities. All the experiments were conducted on NVIDIA A100/H100 GPUs, with each trial involving a hyperparameter search using the Tree-Structured Parzen Estimator (TPE) algorithm over 200 iterations. The results presented in Table 2 and Figure 3 are obtained using 5 different random seeds (selected from 0-4) with the optimal hyperparameter configuration. The training process is capped at 50 epochs, with an early stopping mechanism that terminates training if no accuracy improvement is observed for 10 consecutive epochs. To maintain consistency with the hyperparameter search settings, we employ a two-phase learning rate schedule: during the first 25 epochs, the weight learning rate follows a warmup-cosine-annealing schedule as previously described, after which it remains fixed at the final learning rate of the scheduler. For the results shown in Figure 6, 7 and 8, we utilize a single random seed with the optimal hyperparameters, setting the maximum training epochs to 50 without implementing early

*Table 7.* Comparison of the training times (seconds per epoch) of BP against PCNs on different architectures with CIFAR10

| Task | BP | PC | PC + S + F | iPC | iPC + S |
|---|---|---|---|---|---|
| VGG5 (T = 5) | $1.16^{\pm0.02}$ | $1.56^{\pm0.01}$ | $1.55^{\pm0.01}$ | $1.94^{\pm0.01}$ | $1.94^{\pm0.01}$ |
| VGG7 (T = 7) | $1.29^{\pm0.02}$ | $2.20^{\pm0.01}$ | $2.20^{\pm0.01}$ | $2.94^{\pm0.01}$ | $2.95^{\pm0.01}$ |
| VGG10 (T = 10) | $1.90^{\pm0.01}$ | $5.08^{\pm0.02}$ | $5.06^{\pm0.02}$ | $7.00^{\pm0.05}$ | $7.01^{\pm0.03}$ |
| ResNet10 (T = 10) | $1.75^{\pm0.01}$ | $5.02^{\pm0.02}$ | $5.10^{\pm0.01}$ | $7.08^{\pm0.05}$ | $7.08^{\pm0.05}$ |
| ResNet18 (T = 18) | $2.78^{\pm0.01}$ | $14.92^{\pm0.01}$ | $15.17^{\pm0.03}$ | $22.14^{\pm0.05}$ | $22.14^{\pm0.01}$ |

stopping. The weight learning rate schedule remains identical to the aforementioned approach.

## E. Computational Complexity

In Table 7, we present the average time required to train one epoch using BP, PC, iPC, iPC with Spiking Precision (iPC + S) and PC with Spiking Precision and Forward Update (PC + S + F) across various tasks on a single H100 GPU. To eliminate the overhead associated with loading datasets into memory, we began timing from the fifth epoch onward, calculating the average duration across five consecutive epochs. We repeated this measurement process five times and report the mean and standard deviation of these five experimental runs. It is worth noting that the reported times for predictive coding suffer from an implementation bottleneck: despite the possibility of updating all the neural activities in parallel, our library does not allow that. This largely slows down our models when trained on deep architectures.

**Results.** The results in Table 7 lead to several key observations. First, our proposed methods, PC+S+F and iPC+S, exhibit training times nearly identical to their respective baselines, PC and iPC. This demonstrates that the proposed spiking precision and forward update mechanisms do not introduce a significant computational overhead.

The table also shows that iPC is consistently slower than PC. This performance difference stems from their distinct weight update strategies. PC first runs the inference learning for T timesteps to allow the neural activities ($x$) to converge and then performs a single weight update at the end. In contrast, iPC performs weight updates within each of the T inference steps, alongside the neural activity updates. This approach results in T separate weight updates for every layers instead of one, is the direct cause of its higher computational cost compared to PC.

Finally, all PC models are slower than BP, with this ratio increasing as the number of model layers increases, mostly for the bottleneck just described. While the forward pass is computationally identical for both BP and PC, their backward/update passes differ fundamentally. BP computes gradients in a single backward pass. In contrast, PC perform an iterative inference process to update neural activities by minimizing a global prediction error. This process runs for a fixed number of T. The computational complexity of a single BP epoch is proportional to the number of layers, L. For PC, each of the T inference steps involves computations across all layers, making their complexity roughly proportional to T×L. Since optimal performance often requires T to be equal to or greater than the network depth L, the computational cost for PC naturally scales more rapidly with deeper architectures, despite this not being as much of a bottleneck as the full parallelization of the operations. Thus, We expect predictive coding networks to maintain computational efficiency across larger model architectures while offering substantial performance advantages when implemented on specialized analog neuromorphic hardware.

## F. Ablation Study

In this section, we conduct ablation studies to evaluate the individual and synergistic effects of our proposed components. By systematically isolating each mechanism, we quantify its specific contribution to the overall performance of the model.

### F.1. Full Factorial Ablation

To enable clear attribution of gains to individual components, we conducted a full factorial ablation across all proposed methods: Spiking Precision (S), Forward Update (F), and BatchNorm Freezing (BF). Table. 9 summarizes which components are active in each configuration. The results on CIFAR10 and CIFAR100 are reported in Table. 8. Figure. 5 provides a radar

*Table 8.* Ablation study results on CIFAR10 and CIFAR100. Test accuracies reported as mean $\pm$ std over 5 runs.

| Dataset | Configuration | VGG5 | VGG7 | VGG10 | ResNet10 | ResNet18 |
|---------|---------------|------|------|-------|----------|----------|
| CIFAR10 | PC | $87.98^{\pm0.11}$ | $84.62^{\pm0.10}$ | $72.75^{\pm6.03}$ | $69.85^{\pm2.12}$ | $15.63^{\pm7.22}$ |
| | PC + BF | $87.77^{\pm0.14}$ | $80.62^{\pm0.14}$ | $76.22^{\pm0.43}$ | $64.23^{\pm0.27}$ | $15.21^{\pm0.21}$ |
| | PC + S | $88.06^{\pm0.16}$ | $82.16^{\pm0.14}$ | $78.09^{\pm0.61}$ | $80.61^{\pm0.20}$ | $80.07^{\pm0.21}$ |
| | PC + F | $88.79^{\pm0.04}$ | $87.43^{\pm0.30}$ | $87.33^{\pm0.14}$ | $66.94^{\pm0.76}$ | $25.50^{\pm3.12}$ |
| | PC + S + F | $89.45^{\pm0.18}$ | $90.08^{\pm0.21}$ | $92.07^{\pm0.10}$ | $92.04^{\pm0.04}$ | $91.91^{\pm0.09}$ |
| | PC + S + BF | $88.53^{\pm0.07}$ | $86.51^{\pm0.21}$ | $83.17^{\pm0.07}$ | $81.23^{\pm0.21}$ | $81.10^{\pm0.10}$ |
| | PC + F + BF | $87.34^{\pm0.13}$ | $85.91^{\pm0.09}$ | $79.43^{\pm0.20}$ | $62.35^{\pm0.53}$ | $20.10^{\pm2.10}$ |
| | PC + S + F + BF | $89.30^{\pm0.13}$ | $90.89^{\pm0.04}$ | $93.27^{\pm0.10}$ | $92.47^{\pm0.01}$ | $91.93^{\pm0.14}$ |
| | iPC | $85.51^{\pm0.12}$ | $80.15^{\pm0.18}$ | $63.83^{\pm0.33}$ | $62.34^{\pm0.27}$ | $21.90^{\pm1.51}$ |
| | iPC + BF | $86.01^{\pm0.10}$ | $65.34^{\pm0.20}$ | $61.86^{\pm0.18}$ | $58.80^{\pm0.12}$ | $17.80^{\pm0.10}$ |
| | iPC + S | $89.32^{\pm0.13}$ | $90.25^{\pm0.06}$ | $93.03^{\pm0.18}$ | $92.39^{\pm0.04}$ | $91.96^{\pm0.07}$ |
| | iPC + S + BF | $89.73^{\pm0.06}$ | $91.12^{\pm0.08}$ | $89.36^{\pm0.29}$ | $91.40^{\pm0.01}$ | $90.70^{\pm0.10}$ |
| CIFAR100 | PC | $60.00^{\pm0.19}$ | $56.80^{\pm0.14}$ | $45.86^{\pm1.70}$ | $27.62^{\pm3.03}$ | $1.59^{\pm0.02}$ |
| | PC + BF | $61.84^{\pm0.18}$ | $55.57^{\pm0.14}$ | $50.76^{\pm0.37}$ | $41.51^{\pm0.32}$ | $1.23^{\pm0.30}$ |
| | PC + S | $59.18^{\pm0.20}$ | $56.98^{\pm0.19}$ | $51.56^{\pm0.16}$ | $50.23^{\pm0.20}$ | $22.92^{\pm0.15}$ |
| | PC + F | $65.34^{\pm0.07}$ | $64.50^{\pm0.14}$ | $61.69^{\pm0.79}$ | $39.89^{\pm0.90}$ | $3.42^{\pm0.10}$ |
| | PC + S + F | $66.49^{\pm0.15}$ | $66.34^{\pm0.22}$ | $69.08^{\pm0.08}$ | $68.99^{\pm0.18}$ | $70.81^{\pm0.08}$ |
| | PC + S + BF | $60.34^{\pm0.28}$ | $55.74^{\pm0.15}$ | $56.24^{\pm0.37}$ | $53.17^{\pm0.20}$ | $24.12^{\pm0.20}$ |
| | PC + F + BF | $64.69^{\pm0.17}$ | $63.52^{\pm0.32}$ | $60.33^{\pm0.30}$ | $38.77^{\pm0.70}$ | $2.26^{\pm0.10}$ |
| | PC + S + F + BF | $67.16^{\pm0.16}$ | $67.71^{\pm0.10}$ | $72.02^{\pm0.12}$ | $71.30^{\pm0.21}$ | $70.90^{\pm0.18}$ |
| | iPC | $56.07^{\pm0.16}$ | $43.99^{\pm0.30}$ | $21.37^{\pm0.37}$ | $22.91^{\pm0.23}$ | $1.53^{\pm0.06}$ |
| | iPC + BF | $56.01^{\pm0.19}$ | $41.08^{\pm0.19}$ | $31.99^{\pm0.17}$ | $1.48^{\pm0.05}$ | $1.56^{\pm0.24}$ |
| | iPC + S | $65.54^{\pm0.62}$ | $65.76^{\pm0.12}$ | $69.84^{\pm0.17}$ | $70.02^{\pm0.24}$ | $70.38^{\pm0.20}$ |
| | iPC + S + BF | $66.91^{\pm0.12}$ | $67.10^{\pm0.12}$ | $64.15^{\pm0.30}$ | $69.95^{\pm0.23}$ | $69.23^{\pm0.10}$ |

chart visualizing relative component contributions across architectures.

### F.2. Spiking Precision

#### F.2.1. COMPARISON OF DIFFERENT PRECISION SCHEDULES

**Dynamic Precision.** Our central hypothesis is that mitigating the energy imbalance in deep networks requires a potent and precisely timed signal amplification. This amplification should occur at the moment the error information, propagating from the output layer, first arrives at a given hidden layer. To test this hypothesis, we designed and compared several dynamic precision schedules.

Based on this perspective, in addition to Spiking Precision, we designed a Decaying Precision schedule, which offers a slightly smoother, yet still powerful, amplification profile. The formula for Decaying Precision is as follows:

$$\Sigma_t^l = \begin{cases} \frac{\sum_{j=0}^{T-L+l} e^{-k \cdot j}}{e^{-k \cdot (l-L+t)}}, & \text{when } l \geq L - t, \\ 1, & \text{when } l < L - t. \end{cases} \quad (19)$$

Here, the numerator sum serves as a normalization term that ensures that the sum of the layer-wise precisions over time is equal to one, that is, $\sum_{t=1}^{T}(\Sigma_t^l)^{-1} = 1$. The denominator $e^{-k \cdot (l-L+t)}$ allows lower layers to receive larger weights when activated ($l \geq L - t$), thereby helping to achieve a more balanced energy distribution during the inference phase, k is a hyperparameter that controls the strength of this balancing effect, the search range is $[1.0, 1.5, 2.0]$. It also ensures that each layer experiences a significant boost in precision precisely when the energy from the output first reaches that layer ($l = L - t$). When $l < L - t$, we set $\Sigma_t^l = 1$.

As shown in Table 10, both Decaying and Spiking Precision schedules offer improvements over the baseline PC, particularly in deeper models like VGG10. However, a clear pattern emerges when we analyze their effectiveness in relation to network

*Table 9.* Configuration for the ablation study.

| Configuration | Spiking Prec. (S) | Forward Upd. (F) | BN Freezing | Aux Neurons (For ResNet) |
|---|---|---|---|---|
| **Predictive Coding (PC) Variants** | | | | |
| PC | | | | |
| PC + BF | | | ✓ | |
| PC + S | ✓ | | | ✓ |
| PC + F | | ✓ | | ✓ |
| PC + S + F | ✓ | ✓ | | ✓ |
| PC + S + BF | ✓ | | ✓ | ✓ |
| PC + F + BF | | ✓ | ✓ | ✓ |
| PC + S + F + BF | ✓ | ✓ | ✓ | ✓ |
| **Incremental Predictive Coding (iPC) Variants** | | | | |
| iPC | | ✓* | | |
| iPC + BF | | ✓* | ✓ | |
| iPC + S | ✓ | ✓* | | ✓ |
| iPC + S + BF | ✓ | ✓* | ✓ | ✓ |

\* By updating weights and neural activities simultaneously, iPC inherently captures the Forward Update mechanism while maintaining biological plausibility.

depth. In shallower models like VGG5 and VGG7, the performance of Decaying Precision is comparable to that of Spiking Precision. This suggests that when the signal path is short, a moderately amplification is sufficient.

As shown in the Tab. 11, when model depth increases to VGG15 with TinyImageNet task, a noticeable performance gap appears, with Spiking Precision consistently outperforming Decaying Precision. This finding strongly supports our core hypothesis: the exponential signal attenuation in deeper networks necessitates a correspondingly sharp and powerful counteracting signal. The abrupt, targeted amplification of Spiking Precision is more effective at preserving the integrity of the error signal across many layers than the smoother profile of Decaying Precision. Consequently, for all subsequent experiments reported in the main body of this paper, we exclusively utilized the superior Spiking Precision schedule. This investigation also opens exciting avenues for future work, such as exploring hybrid schedules that might combine the strengths of different amplification profiles.

**Fixed Layer-Specific Precision.** A natural alternative to dynamic precision is a fixed, layer-specific precision that scales linearly across the network. We tested this by setting the bottom-layer precision to $a$ and the top-layer precision to $b$, with intermediate layers linearly interpolated, treating $a$ and $b$ as hyperparameters and performing a search over them.

The best configuration was $a = 0.9$ (bottom) and $b = 0.05$ (top), achieving $88.21\%$ on VGG10/CIFAR10 (Tab. 12). This is better than the PC+F baseline ($87.33\%$) because it partially mitigates the spatial energy imbalance, but well below Spiking Precision ($92.07\%$). The fixed scheme helps spatially but misses the *temporal* aspect: Spiking Precision concentrates the gain at the exact moment the error signal first arrives at each layer, rather than applying a static rescaling across all time steps.

F.2.2. ENERGY PROPAGATION WITH PRECISION

We observed that removing the decaying/spiking precision module consistently leads to performance degradation. This effect is particularly evident in deeper models like VGG7 and VGG10, where its absence causes a significant imbalance in the energy distribution across layers. For instance, as shown in Figure 7, the energy proportion of the first layer in the VGG7 model with spiking precision is approximately $10^{-6}$. Without this precision term, the proportion plummets to $10^{-18}$. A comparison of the layer-wise energy distributions (Figure 8) confirms that our proposed precision methods effectively rebalance energy propagation. By increasing the energy in the initial layers by several orders of magnitude, these methods rectify the imbalance, which contributes directly to improved model performance.

Furthermore, the degree of this energy rebalancing correlates with the performance difference between the Decaying

*Table 10.* Test accuracies of the different algorithms across architectures and datasets.

| Dataset | Algorithm | VGG5 | VGG7 | VGG10 | VGG5BF | VGG7BF | VGG10BF |
|---|---|---|---|---|---|---|---|
| | BP | $89.43^{\pm0.12}$ | $89.91^{\pm0.12}$ | $\mathbf{92.21^{\pm0.08}}$ | $90.01^{\pm0.15}$ | $91.32^{\pm0.14}$ | $92.68^{\pm0.10}$ |
| | PC | $87.98^{\pm0.11}$ | $84.62^{\pm0.10}$ | $72.75^{\pm6.03}$ | $87.77^{\pm0.14}$ | $80.62^{\pm0.14}$ | $76.22^{\pm0.43}$ |
| CIFAR10 | Decaying Precision (D) | $87.91^{\pm0.22}$ | $81.14^{\pm0.19}$ | $84.87^{\pm0.19}$ | $88.55^{\pm0.09}$ | $80.00^{\pm0.10}$ | $81.68^{\pm0.18}$ |
| | Spiking Precision (S) | $88.06^{\pm0.16}$ | $82.16^{\pm0.14}$ | $78.09^{\pm0.61}$ | $88.53^{\pm0.07}$ | $86.51^{\pm0.21}$ | $83.17^{\pm0.07}$ |
| | PC+D+F | $89.32^{\pm0.14}$ | $89.34^{\pm0.09}$ | $89.43^{\pm0.18}$ | $\mathbf{90.37^{\pm0.13}}$ | $\mathbf{91.48^{\pm0.12}}$ | $91.46^{\pm0.12}$ |
| | PC+S+F | $\mathbf{89.45^{\pm0.18}}$ | $\mathbf{90.08^{\pm0.21}}$ | $92.07^{\pm0.10}$ | $89.30^{\pm0.13}$ | $90.89^{\pm0.04}$ | $\mathbf{93.27^{\pm0.10}}$ |
| | BP | $66.28^{\pm0.23}$ | $65.36^{\pm0.15}$ | $\mathbf{69.35^{\pm0.16}}$ | $67.39^{\pm0.25}$ | $67.67^{\pm0.11}$ | $71.25^{\pm0.21}$ |
| | PC | $60.00^{\pm0.19}$ | $56.80^{\pm0.14}$ | $45.86^{\pm1.70}$ | $61.84^{\pm0.18}$ | $55.57^{\pm0.14}$ | $50.76^{\pm0.37}$ |
| CIFAR100 (Top-1) | Decaying Precision (D) | $57.76^{\pm0.33}$ | $45.05^{\pm0.37}$ | $55.66^{\pm0.88}$ | $66.05^{\pm0.12}$ | $51.11^{\pm0.32}$ | $53.27^{\pm0.48}$ |
| | Spiking Precision (S) | $59.18^{\pm0.20}$ | $56.98^{\pm0.19}$ | $51.56^{\pm0.16}$ | $60.34^{\pm0.28}$ | $55.74^{\pm0.15}$ | $56.24^{\pm0.37}$ |
| | PC+D+F | $66.10^{\pm0.09}$ | $64.86^{\pm0.10}$ | $66.54^{\pm0.12}$ | $\mathbf{67.56^{\pm0.25}}$ | $67.27^{\pm0.21}$ | $69.81^{\pm0.22}$ |
| | PC+S+F | $\mathbf{66.49^{\pm0.15}}$ | $\mathbf{66.34^{\pm0.22}}$ | $69.08^{\pm0.08}$ | $67.16^{\pm0.16}$ | $\mathbf{67.71^{\pm0.10}}$ | $\mathbf{72.02^{\pm0.12}}$ |
| | BP | $85.85^{\pm0.27}$ | $84.41^{\pm0.26}$ | $\mathbf{88.74^{\pm0.08}}$ | $89.56^{\pm0.08}$ | $\mathbf{90.05^{\pm0.13}}$ | $92.10^{\pm0.12}$ |
| | PC | $84.97^{\pm0.19}$ | $83.00^{\pm0.09}$ | $74.61^{\pm1.08}$ | $86.53^{\pm0.15}$ | $82.07^{\pm0.35}$ | $78.68^{\pm0.27}$ |
| CIFAR100 (Top-5) | Decaying Precision (D) | $81.59^{\pm0.13}$ | $74.00^{\pm0.30}$ | $83.13^{\pm0.74}$ | $88.82^{\pm0.07}$ | $78.93^{\pm0.29}$ | $81.16^{\pm0.36}$ |
| | Spiking Precision (S) | $84.58^{\pm0.12}$ | $83.61^{\pm0.15}$ | $78.62^{\pm0.15}$ | $85.86^{\pm0.10}$ | $82.64^{\pm0.14}$ | $83.44^{\pm0.21}$ |
| | PC+D+F | $85.85^{\pm0.10}$ | $83.80^{\pm0.20}$ | $86.10^{\pm0.21}$ | $\mathbf{89.84^{\pm0.17}}$ | $89.74^{\pm0.12}$ | $91.24^{\pm0.07}$ |
| | PC+S+F | $\mathbf{86.36^{\pm0.11}}$ | $\mathbf{84.53^{\pm0.15}}$ | $86.84^{\pm0.07}$ | $89.57^{\pm0.09}$ | $89.62^{\pm0.18}$ | $92.10^{\pm0.10}$ |

*Table 11.* Test accuracies of the different algorithms on Tiny ImageNet.

| Algorithm | Top-1 Accuracy | | Top-5 Accuracy | |
|---|---|---|---|---|
| | VGG15 | VGG15BF | VGG15 | VGG15BF |
| PC+S+Forward Update | $42.51^{\pm0.18}$ | $53.04^{\pm0.36}$ | $66.22^{\pm0.18}$ | $76.64^{\pm0.23}$ |
| PC | $22.95^{\pm1.50}$ | $22.91^{\pm0.61}$ | $47.04^{\pm2.04}$ | $45.64^{\pm0.63}$ |
| Decaying Precision (D) | $27.29^{\pm0.24}$ | $22.05^{\pm0.16}$ | $54.18^{\pm0.37}$ | $45.22^{\pm0.23}$ |
| Spiking Precision (S) | $18.36^{\pm0.36}$ | $17.95^{\pm0.14}$ | $39.24^{\pm0.31}$ | $39.87^{\pm0.51}$ |
| PC+D+Forward Update | $21.95^{\pm0.20}$ | $30.83^{\pm0.77}$ | $45.15^{\pm0.23}$ | $56.06^{\pm0.85}$ |

and Spiking Precision variants. In the VGG5 and VGG7 models, where the accuracy gap between the two methods is minimal, the difference in their first-layer energy distributions is also small. However, in VGG10, where Spiking Precision significantly outperforms Decaying Precision, the energy gap is far more pronounced. Specifically, the first layer's energy proportion is approximately $10^{-10}$ for Decaying Precision, whereas Spiking Precision elevates it to $10^{-4}$, highlighting a clear link between balanced energy propagation and model accuracy.

### F.3. Forward Update (F)

F.3.1. NEURAL ACTIVITY DIVERGENCY QUANTIFICATION

To quantify the neural activity divergence that Forward Update aims to solve, we conducted a new experiment on a VGG10 model trained on CIFAR-10. We measured the Mean Squared Error between the initial and final neural states, $MSE(x_0^l, x_T^l)$, at each weight update. We then calculated the ratio of the square root of this divergence to the energy used for the weight update in that layer. We term this metric the "Gap Ratio". As shown in Table 13, in the model trained without Forward Update, the Gap Ratio in the final layer (L10) is extremely large and unstable across epochs, indicating that the neural activity divergence completely dominates the weight update signal. This supports our hypothesis that this divergence causes errors to accumulate in the final layers, destabilizing learning. In contrast, the model trained with Forward Update (Table 14) shows a dramatically reduced and stable Gap Ratio in the final layer.

To further isolate the effect of Forward Update, we performed an additional ablation where FU was applied only to the final three layers during the weight update, without any additional hyperparameter tuning. The results in Table 15 show that even this targeted application of Forward Update yields significant improvements in both training stability and final accuracy compared to the baseline. This reinforces that addressing the neural activity divergence in the deepest layers is a critical

*Table 12.* Comparison of test accuracy across different precision strategies on VGG10/CIFAR10.

| Method | Test Accuracy (%) |
|---|---|
| PC+F (Baseline) | $87.33^{\pm 0.14}$ |
| PC + Fixed Layer-specific Precision + F | $88.21^{\pm 0.12}$ |
| PC + Spiking Precision + F | $\mathbf{92.07^{\pm 0.10}}$ |

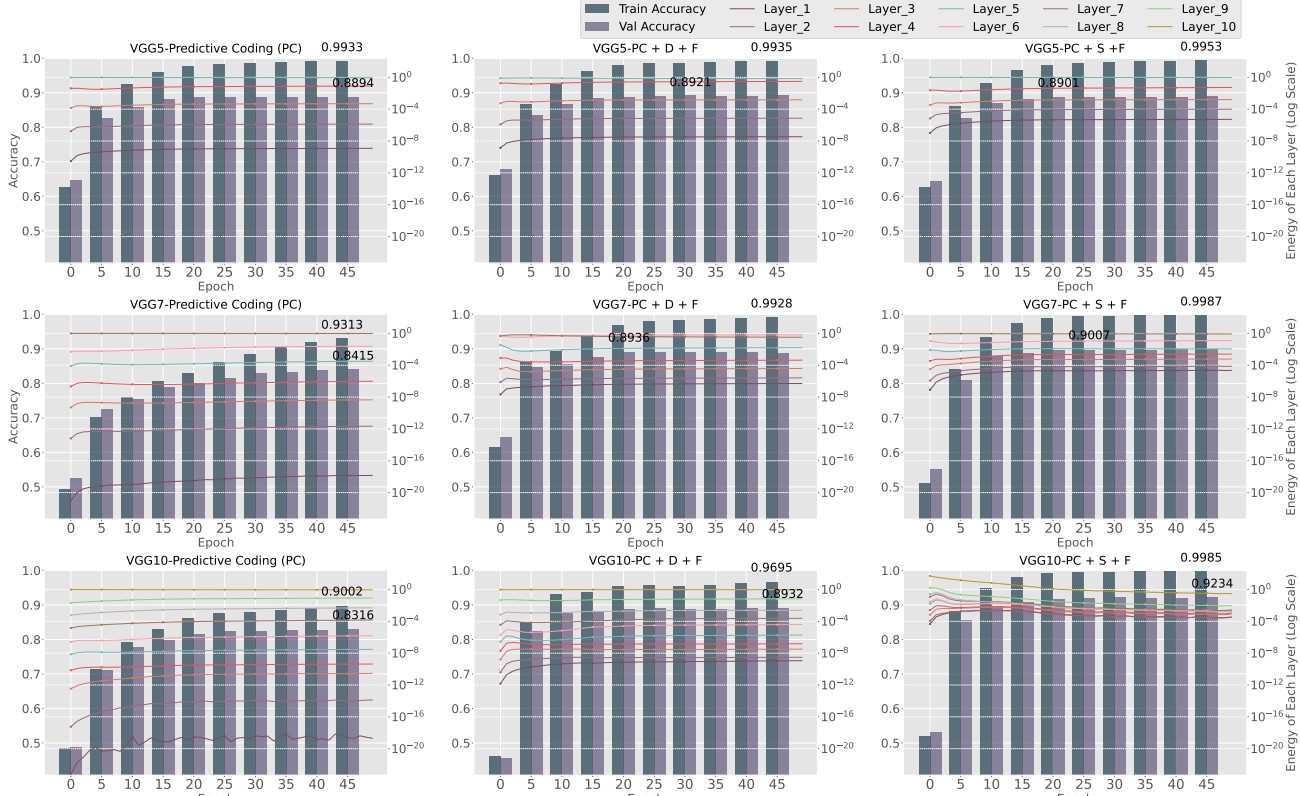

*Figure 7.* Layer-wise Energy Distribution and Accuracy Comparison between PC and Decaying Precision/Spiking Precision with Forward Update in VGG5, VGG7 and VGG10 on the CIFAR10 dataset. The colored lines represent the total energy of the individual layers of the model. The vertical lines represent the train and test accuracies of the model.

*Table 13.* Gap Ratio on VGG10/CIFAR10 trained without Forward Update.

| w/o F | L1 | L2 | L3 | L4 | L5 | L6 | L7 | L8 | L9 | L10 | Train Acc. | Test Acc. |
|---|---|---|---|---|---|---|---|---|---|---|---|---|
| Epoch 5 | 0.0 | 99.49 | 98.98 | 97.08 | 106.78 | 113.52 | 110.77 | 468.9 | 566.52 | 7065.72 | 73.75 | 74.83 |
| Epoch 15 | 0.0 | 99.51 | 99.4 | 97.2 | 98.95 | 105.62 | 132.44 | 108.39 | 101.0 | 3807.9 | 51.86 | 51.9 |
| Epoch 25 | 0.0 | 99.25 | 97.94 | 98.22 | 89.95 | 90.48 | 116.55 | 113.2 | 102.21 | 4716.88 | 61.21 | 62.3 |
| Epoch 35 | 0.0 | 99.41 | 98.24 | 96.97 | 79.12 | 95.99 | 122.38 | 106.3 | 104.07 | 6498.41 | 60.31 | 60.14 |
| Epoch 45 | 0.0 | 98.23 | 97.94 | 97.36 | 80.84 | 90.42 | 116.96 | 108.26 | 106.42 | 5362.02 | 59.87 | 61.42 |

factor for success.

### F.3.2. MODEL ROBUSTNESS ANALYSIS

Predictive Coding (PC) networks are often noted for their inherent robustness compared to networks trained with Backprop-agation (Salvatori et al., 2021). Unlike our precision-weighting mechanisms, the Forward Update (F) method alters the core computational diagram of PC. Therefore, we conducted experiments to investigate whether this modification adversely

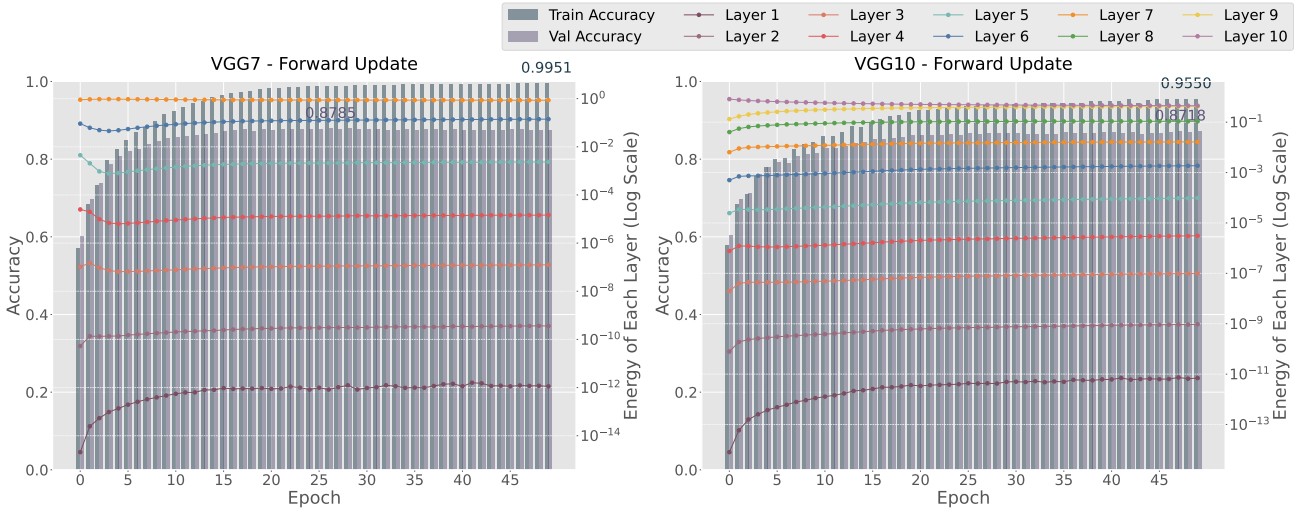

*Figure 8.* Layer-wise Energy Distribution and Accuracy of Forward Update in VGG7 and VGG10 on the CIFAR10 dataset. The colored lines represent the total energy of the individual layers of the model. The vertical lines represent the train and test accuracies of the model.

*Table 14.* Gap Ratio on VGG10/CIFAR10 trained with Forward Update.

| with F | L1 | L2 | L3 | L4 | L5 | L6 | L7 | L8 | L9 | L10 | Train Acc. | Test Acc. |
|---|---|---|---|---|---|---|---|---|---|---|---|---|
| Epoch 5 | 0.0 | 292.58 | 112.77 | 224.2 | 120.89 | 346.42 | 143.95 | 248.32 | 100.78 | 5.14 | 88.27 | 85.66 |
| Epoch 15 | 0.0 | 327.43 | 132.66 | 265.24 | 151.09 | 409.26 | 264.34 | 545.12 | 117.6 | 8.27 | 97.97 | 90.99 |
| Epoch 25 | 0.0 | 323.0 | 135.82 | 256.2 | 149.77 | 407.2 | 269.14 | 619.95 | 115.07 | 9.43 | 99.55 | 92.08 |
| Epoch 35 | 0.0 | 322.42 | 138.05 | 258.38 | 148.22 | 415.02 | 292.98 | 681.21 | 115.12 | 10.06 | 99.73 | 92.04 |
| Epoch 45 | 0.0 | 323.25 | 135.72 | 261.67 | 149.45 | 417.04 | 312.4 | 731.19 | 115.87 | 10.54 | 99.83 | 92.22 |

*Table 15.* Gap Ratio on VGG10/CIFAR10 when applying Forward Update only to the last 3 layers (L8, L9, L10).

| with F (3 Layers) | L1 | L2 | L3 | L4 | L5 | L6 | L7 | L8 | L9 | L10 | Train Acc. | Test Acc. |
|---|---|---|---|---|---|---|---|---|---|---|---|---|
| Epoch 5 | 0.0 | 96.21 | 57.22 | 50.78 | 32.03 | 53.59 | 30.16 | 49.77 | 66.91 | 42.19 | 72.75 | 74.13 |
| Epoch 15 | 0.0 | 95.51 | 57.52 | 49.76 | 31.65 | 56.92 | 32.96 | 40.87 | 70.08 | 62.02 | 79.23 | 78.56 |
| Epoch 25 | 0.0 | 95.58 | 59.17 | 46.63 | 16.04 | 91.18 | 34.19 | 39.63 | 73.17 | 67.89 | 81.4 | 80.14 |
| Epoch 35 | 0.0 | 95.44 | 56.84 | 29.72 | 9.65 | 94.43 | 35.13 | 39.37 | 72.12 | 70.17 | 80.91 | 79.97 |
| Epoch 45 | 0.0 | 95.51 | 56.88 | 37.85 | 15.11 | 93.49 | 34.8 | 38.59 | 71.82 | 72.16 | 80.02 | 79.4 |

affects model robustness.

Our first experiment provides a fair comparison on a VGG5 model, where standard PC and our PC+F variant achieve similar baseline accuracies on clean data. We trained both models on CIFAR-10 and evaluated their calibration under six types of data corruption across five levels of intensity. The range of the Adaptive Expected Calibration Error (adaECE) is reported in Table 16. Note that to ensure a meaningful ECE calculation, we scale the output logits before applying the softmax function. The results indicate that our PC+F model maintains a robustness profile that is comparable to, and at higher corruption levels, superior to that of standard PC.

Furthermore, to evaluate our methods on deeper models, we compared our PC+S+F model against the BP baseline on the VGG10 architecture. As shown in Table 17, our method exhibits a robustness profile that is highly comparable to backpropagation across all tested corruption intensities. Taken together, these experiments demonstrate that our proposed methods not only enable the training of deeper PCNs but do so while preserving the desirable property of model robustness.

*Table 16.* AdaECE range at different levels of corruption using VGG5.

| Corruption Level | PC+F | PC |
|---|---|---|
| 0.1 | [0.091665, 0.327372] | [0.037817, 0.388640] |
| 0.2 | [0.089805, 0.349025] | [0.036528, 0.406428] |
| 0.3 | [0.034430, 0.363157] | [0.035065, 0.419023] |
| 0.4 | [0.021906, 0.375255] | [0.029735, 0.428372] |
| 0.5 | [0.026589, 0.383087] | [0.033315, 0.434768] |

*Table 17.* AdaECE range at different levels of corruption using VGG10.

| Corruption Level | BP | PC+S+F |
|---|---|---|
| 0.1 | [0.039071, 0.485300] | [0.039269, 0.486342] |
| 0.2 | [0.043376, 0.490541] | [0.045973, 0.486978] |
| 0.3 | [0.043943, 0.494404] | [0.047871, 0.486432] |
| 0.4 | [0.067549, 0.496167] | [0.077340, 0.486966] |
| 0.5 | [0.132682, 0.499181] | [0.143458, 0.486497] |

### F.3.3. ON THE BIOLOGICAL PLAUSIBILITY OF FORWARD UPDATE

A core motivation for using PCNs over backpropagation is their biological plausibility — particularly their use of local learning rules and temporally local computations. The Forward Update mechanism, while effective, seems to introduce non-locality in time by requiring each synapse to store its initial feedforward activity $\mu_0^l$ until convergence.

However, the description of *Forward Update* in our manuscript was chosen for conceptual clarity; it is not a fundamental requirement of the method. In practice, the initial feed-forward state can be re-computed through a temporally local and biologically plausible process. This is achieved by introducing a "free relaxation" phase after the inference learning and before the weight update. In this phase, the label clamp is removed, and the network settles to a new equilibrium with only the sensory input clamped, just as in Equilibrium Propagation (Ernoult et al., 2020) before the nudging phase.

In this setting, the network naturally converges to the state corresponding to its feed-forward prediction (Frieder & Lukasiewicz, 2022). Since the weights have not yet been updated, this re-computed state is identical to the $\mu_0^l$ in our formulation. This eliminates the need for long-term storage and resolves the concern of temporal non-locality. The primary contribution of our *Forward Update* method is that it identifies and solves a critical failure mode in deep predictive coding networks: the accumulation of errors caused by the divergence of neural activities from their initial predictions. While the biologically plausible implementation we describe requires extra computation, future work can focus on developing mechanisms that are both fully plausible and computationally efficient for PC.

### F.4. Ablation study on Forward Update and Spiking precision

To dissect the individual and combined contributions of our primary algorithmic modifications, Spiking Precision (S) and Forward Update (F), we conducted a detailed ablation study, the results of which are presented in Table 18. This analysis systematically evaluates each component's impact on both standard PC and iPC across various architectures and datasets, excluding the effects of BatchNorm Freezing to isolate the core mechanisms.

The results reveal a clear and complementary relationship between Spiking Precision and Forward Update for standard PC. When applied in isolation, Forward Update (PC+F) significantly improves performance on VGG-style architectures, stabilizing training and preventing the sharp accuracy degradation seen in the baseline PC as depth increases. For instance, on CIFAR10, PC+F maintains an accuracy of around 87% on VGG10, whereas the baseline PC drops to 72.75%. However, Forward Update alone is insufficient for training deep residual networks; its performance on ResNet18 is only marginally better than the baseline, failing to overcome the catastrophic failure. This suggests that while F effectively mitigates weight update divergence, it does not solve the underlying problem of energy imbalance in architectures with skip connections.

Conversely, Spiking Precision alone (PC+S) offers a substantial improvement on ResNet models, preventing the complete collapse of training. On ResNet18 with CIFAR10, it achieves an accuracy of 80.07%, a dramatic recovery from the baseline's 15.63%. This confirms its crucial role in rebalancing energy and ensuring a viable error signal reaches the early layers in

*Table 18.* Test accuracies of different algorithms without BatchNorm Freeze across datasets and architectures.

| Dataset | Algorithm | VGG5 | VGG7 | VGG10 | ResNet10 | ResNet18 |
|---|---|---|---|---|---|---|
| CIFAR10 | BP | $89.43^{\pm 0.12}$ | $89.91^{\pm 0.12}$ | $92.21^{\pm 0.08}$ | $92.21^{\pm 0.20}$ | $\mathbf{92.32^{\pm 0.22}}$ |
| | PC | $87.98^{\pm 0.11}$ | $84.62^{\pm 0.10}$ | $72.75^{\pm 6.03}$ | $69.85^{\pm 2.12}$ | $15.63^{\pm 7.22}$ |
| | iPC | $85.51^{\pm 0.12}$ | $80.15^{\pm 0.18}$ | $63.83^{\pm 0.33}$ | $62.34^{\pm 0.27}$ | $21.90^{\pm 1.51}$ |
| | PC+Spiking Precision (S) | $88.06^{\pm 0.16}$ | $82.16^{\pm 0.14}$ | $78.09^{\pm 0.61}$ | $80.61^{\pm 0.20}$ | $80.07^{\pm 0.21}$ |
| | PC+Forward Update (F) | $88.79^{\pm 0.04}$ | $87.43^{\pm 0.30}$ | $87.33^{\pm 0.14}$ | $66.94^{\pm 0.76}$ | $25.50^{\pm 3.12}$ |
| | iPC+S | $89.32^{\pm 0.13}$ | $\mathbf{90.25^{\pm 0.06}}$ | $\mathbf{93.03^{\pm 0.18}}$ | $\mathbf{92.39^{\pm 0.04}}$ | $91.96^{\pm 0.07}$ |
| | PC+S+F | $\mathbf{89.45^{\pm 0.18}}$ | $90.08^{\pm 0.21}$ | $92.07^{\pm 0.10}$ | $92.04^{\pm 0.04}$ | $91.91^{\pm 0.09}$ |
| CIFAR100 (Top-1) | BP | $66.28^{\pm 0.23}$ | $65.36^{\pm 0.15}$ | $69.35^{\pm 0.16}$ | $69.23^{\pm 0.09}$ | $\mathbf{71.46^{\pm 0.12}}$ |
| | PC | $60.00^{\pm 0.19}$ | $56.80^{\pm 0.14}$ | $45.86^{\pm 1.70}$ | $27.62^{\pm 3.03}$ | $1.59^{\pm 0.02}$ |
| | iPC | $56.07^{\pm 0.16}$ | $43.99^{\pm 0.30}$ | $21.37^{\pm 0.37}$ | $22.91^{\pm 0.23}$ | $1.53^{\pm 0.06}$ |
| | PC+Spiking Precision (S) | $59.18^{\pm 0.20}$ | $56.98^{\pm 0.19}$ | $51.56^{\pm 0.16}$ | $50.23^{\pm 0.20}$ | $22.92^{\pm 0.15}$ |
| | PC+Forward Update (F) | $65.34^{\pm 0.07}$ | $64.50^{\pm 0.14}$ | $61.69^{\pm 0.79}$ | $39.89^{\pm 0.90}$ | $3.42^{\pm 0.10}$ |
| | iPC+S | $65.54^{\pm 0.62}$ | $65.76^{\pm 0.12}$ | $\mathbf{69.84^{\pm 0.17}}$ | $\mathbf{70.02^{\pm 0.24}}$ | $70.38^{\pm 0.20}$ |
| | PC+S+F | $\mathbf{66.49^{\pm 0.15}}$ | $\mathbf{66.34^{\pm 0.22}}$ | $69.08^{\pm 0.08}$ | $68.99^{\pm 0.18}$ | $70.81^{\pm 0.08}$ |
| CIFAR100 (Top-5) | BP | $85.85^{\pm 0.27}$ | $84.41^{\pm 0.26}$ | $\mathbf{88.74^{\pm 0.08}}$ | $87.75^{\pm 0.10}$ | $89.43^{\pm 0.14}$ |
| | PC | $84.97^{\pm 0.19}$ | $83.00^{\pm 0.09}$ | $74.61^{\pm 1.08}$ | $57.93^{\pm 2.62}$ | $5.89^{\pm 0.12}$ |
| | iPC | $78.91^{\pm 0.23}$ | $73.23^{\pm 0.30}$ | $48.35^{\pm 0.79}$ | $46.41^{\pm 0.31}$ | $6.33^{\pm 0.26}$ |
| | PC+Spiking Precision (S) | $84.58^{\pm 0.12}$ | $83.61^{\pm 0.15}$ | $78.62^{\pm 0.15}$ | $77.84^{\pm 0.23}$ | $53.89^{\pm 0.06}$ |
| | PC+Forward Update (F) | $85.48^{\pm 0.08}$ | $84.05^{\pm 0.07}$ | $76.73^{\pm 0.92}$ | $69.48^{\pm 0.70}$ | $15.25^{\pm 0.04}$ |
| | iPC+S | $85.66^{\pm 0.29}$ | $\mathbf{84.96^{\pm 0.14}}$ | $88.70^{\pm 0.18}$ | $\mathbf{88.90^{\pm 0.21}}$ | $90.05^{\pm 0.20}$ |
| | PC+S+F | $\mathbf{86.36^{\pm 0.11}}$ | $84.53^{\pm 0.15}$ | $88.66^{\pm 0.14}$ | $88.45^{\pm 0.16}$ | $\mathbf{90.47^{\pm 0.11}}$ |

models with skip-connections. However, on its own, it does not elevate performance to the level of backpropagation.

The true strength of our approach is demonstrated when both components are combined. The PC+S+F model consistently achieves performance on par with, and occasionally exceeding, backpropagation across all tested architectures, including the challenging ResNet18. This powerful synergy underscores that both mechanisms are essential: Spiking Precision addresses the signal propagation problem, while Forward Update addresses the update stability problem.

Interestingly, for iPC, the addition of Spiking Precision alone (iPC+S) is sufficient to achieve state-of-the-art performance, rivaling both BP and the fully-equipped PC+S+F. This indicates that the incremental nature of iPC, where weights are updated at every inference timestep, inherently prevents the large divergence between forward neural state and backward neural states that Forward Update is designed to correct. With its continuous adaptation, iPC only requires the energy rebalancing provided by Spiking Precision to successfully train deep architectures.

### F.5. BatchNorm Freezing (BF)

To isolate the precise source of instability when applying BatchNorm (BN) to Predictive Coding Networks (PCNs), we sought to determine whether the problem stems from the learnable affine parameters ($\gamma, \beta$) or from the iterative updating of batch statistics ($\mu_B, \sigma_B^2$) during the inference phase. While the affine parameters can be a source of overfitting in standard training, we hypothesized that the unique, multi-step inference process of PCNs creates a different challenge: the repeated processing of a single mini-batch causes the batch statistics themselves to overfit, destabilizing the network dynamics. Our BatchNorm Freezing (BF) method is designed specifically to solve this issue.

To test this hypothesis, we conducted an ablation study on the VGG10-CIFAR10 task. The results presented in Table 19, provide clear evidence for our claim. Removing the affine parameters from standard BN still had a negligible effect on the performance degradation ($24.31\%$ and $24.28\%$), indicating that these parameters are not the source of the problem. In contrast, freezing the statistics during inference improved performance over the without batch normalization baseline. This confirms that the iterative updates to batch statistics are the primary cause of instability and that our proposed BF method is an effective solution.

*Table 19.* Ablation study on different BatchNorm strategies (VGG10-CIFAR10).

| Method | Test Accuracy (%) |
|---|---|
| Without BN | $92.07 \pm 0.10$ |
| Standard BN | $24.31 \pm 4.51$ |
| BN without affine | $24.28 \pm 3.19$ |
| BF without affine | $93.18 \pm 0.06$ |
| **BF** | $\mathbf{93.27 \pm 0.10}$ |

### F.5.1. ABLATION STUDY ON BATCHNORM FREEZING

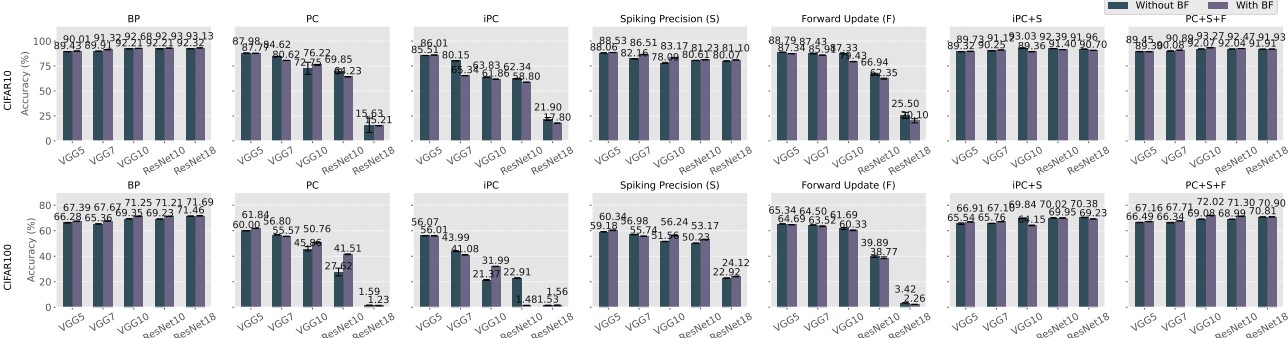

*Figure 9.* Test accuracies of different algorithms on the CIFAR10/100 datasets across models of varying depths, comparing different methods with and without BatchNorm freezing.

Our investigation reveals that BatchNorm Freezing ($BF$) significantly enhances model performance when combined with our precision module and forward update mechanism. As illustrated in Figure 9, the integration of $BF$ with our proposed methods ($PC + S + F$) consistently improved accuracy across all model depths and both CIFAR10 and CIFAR100 datasets. Specifically, the $PC + S + F$ configuration with $BF$ achieved peak performance of 93.27% on CIFAR10, 72.02% on CIFAR100 with the VGG10 architecture and 71.3% on CIFAR100 with the ResNet10 architecture, outperforming even the $BP$ baseline. In contrast, when $BF$ was applied to standard $PC$ or only with forward Update, we observed performance degradation rather than improvement in most cases, the effect of $BF$ was inconsistent and unpredictable across different network depths. These results suggest that the synergy between our proposed components is crucial—$BF$ appears to stabilize the training dynamics specifically when used in conjunction with both our energy balancing mechanisms and forward update. This interaction allows deeper networks to maintain stable gradients throughout the training process, resulting in more robust optimization and ultimately higher classification accuracy.

**The Role of Normalization at Scale on iPC.** The effect of BF on *iPC-based models* is more nuanced. While it improves performance on shallower architectures, a slight performance degradation is observed in deeper models. To explicitly validate this phenomenon on a complex distribution, we conducted an additional experiment scaling up to a VGG-15 on Tiny ImageNet. The results, detailed in Table 20, confirm the findings from our ablation study. We observe that 'PC+S+F' relies on BF to match the BP baseline (**53.04**% vs 53.21%). Conversely, 'iPC+S' performs best *without* any normalization (50.24%), and applying BF actually hurts its performance (48.19%). This discrepancy can be attributed to the nature of iPC's update rule. Because iPC updates weights and neural activities simultaneously, updating BN's running statistics requires an *additional, separate weight update step*. This modification alters the computational graph compared to the baseline model (without BF) that was used for the hyperparameter search, which could explain the suboptimal performance in more complex architectures.

## G. Scaling to Extreme Depth (101-layer ResMLP).

To validate the feasibility of our proposed methods on extremely deep networks, we conducted an experiment using a 101-layer Residual MLP (ResMLP) on the MNIST dataset.

*Table 20.* Test accuracies on Tiny ImageNet.

| % Accuracy | BP | PC+S+F | iPC+S |
|---|---|---|---|
| **VGG-15** | | | |
| Top-1 | $44.52^{\pm0.16}$ | $50.10^{\pm0.16}$ | $\mathbf{50.24^{\pm0.13}}$ |
| Top-5 | $69.28^{\pm0.06}$ | $73.23^{\pm0.18}$ | $\mathbf{75.41^{\pm0.10}}$ |
| **VGG-15-BN/BF** | | | |
| Top-1 | $\mathbf{53.21^{\pm0.39}}$ | $53.04^{\pm0.36}$ | $48.19^{\pm0.71}$ |
| Top-5 | $\mathbf{77.26^{\pm0.17}}$ | $76.64^{\pm0.23}$ | $72.64^{\pm0.20}$ |

Crucially, to demonstrate the robustness and ease of tuning of our framework, we avoided a direct, computationally expensive hyperparameter search on the 101-layer model. Instead, we adopted a hyperparameter transfer strategy. We performed a hyperparameter search on a shallow 8-layer ResMLP proxy and transferred the weight learning rate $\eta_w$ to the 101-layer target model following Depth-$\mu P$ scaling rules (Yang et al., 2024):

$$\eta_w^{(101)} = \eta_w^{(8)} \times \sqrt{\frac{8}{101}}. \tag{20}$$

We kept the inference learning rate $\alpha$ (or $lr_x$) constant and set the inference steps equal to the network depth, i.e., $T = L = 101$.

The results are presented in Table 21. Both PC+S+F and iPC+S+F achieve accuracies comparable to Backpropagation. This result is significant given that we utilized hyperparameters derived solely from a shallow 8-layer proxy. It confirms that our method remains feasible for large-scale deep models and adheres to predictable scaling dynamics without requiring exhaustive retraining.

*Table 21.* Test accuracies (%) on MNIST using a 101-layer ResMLP. Hyperparameters were transferred from an 8-layer proxy model using Depth-$\mu P$ scaling.

| Algorithm | Accuracy (%) |
|---|---|
| BP | 98.40 |
| PC+S+F | 97.13 |
| iPC+S+F | 97.09 |

## H. Scaling to ImageNet-1K.

Scaling to ImageNet-1K is currently constrained by compute, as the sequential operations required by our framework make extensive hyperparameter search prohibitively time-consuming. Because of this, we initially demonstrated scalability across three alternative axes: (1) depth (101-layer ResMLP), (2) label complexity (ImageNet32), and (3) resolution and complexity (Galaxy10). Across all cases, we match BP performance within 1%. We maintain that scaling to full ImageNet is primarily an engineering challenge (e.g., distributed simulation, optimized parallel relaxation) rather than a fundamental algorithmic limitation.

Nevertheless, to provide preliminary evidence of our method's viability at scale, we successfully trained a VGG11BN model on the full ImageNet dataset—even in the absence of hyperparameter tuning—achieving a Top-1 accuracy of 58.86%. Our architecture mirrors the standard VGG11, with the sole modification being the removal of dropout layers. The training process took approximately one week using two NVIDIA A100 GPUs. We trained for 90 epochs with a batch size of 128. The weight learning rate ($lr_w$) followed a cosine decay schedule from $10^{-4}$ to $10^{-5}$, while the state learning rate ($lr_x$) was decayed in a stepwise manner: 0.5 for the first 30 epochs, 0.4 for epochs 30–60, and 0.3 for the final 30 epochs. The BatchNorm momentum was set to 0.1.

## I. Hyperparameter Analysis

### I.1. Hyperparameter Importance

To assess the sensitivity of our approach to different hyperparameters, we conducted a hyperparameter importance analysis using a functional ANOVA (fANOVA) based method. This score quantifies the contribution of each hyperparameter to

the final optimal validation accuracy. The results for CIFAR10 and CIFAR100 on various architectures are summarized in Tables 22. The analysis shows that while the learning rate ($lr_w$) is consistently the most critical parameter, our new parameters (like $\alpha/k$ in Precision) are not overly sensitive and show stable influence across settings, indicating a desirable robustness in our proposed methods.

*Table 22.* Hyperparameter Importance Scores (%) on CIFAR-10 / CIFAR-100. The values are presented as **C10 / C100**.

| Method | Activation | T | $\alpha/k$ | $lr_x$ | $lr_w$ | $momentum_x$ | $weight\_decay_w$ |
|---|---|---|---|---|---|---|---|
| **VGG5** | | | | | | | |
| BP | 0.94 / 9.49 | – / – | – / – | – / – | 98.99 / 90.50 | – / – | 0.07 / 0.01 |
| PC | 5.96 / 3.10 | 1.00 / 3.03 | – / – | 77.29 / 83.65 | 10.98 / 6.56 | 3.51 / 2.55 | 1.26 / 1.12 |
| iPC | 2.20 / 1.14 | 1.15 / 2.18 | – / – | 1.53 / 1.37 | 88.75 / 91.29 | 1.83 / 3.04 | 3.25 / 0.99 |
| iPC+S | 0.85 / 3.37 | 0.52 / 0.62 | 0.22 / 5.74 | 1.13 / 2.92 | 94.02 / 86.36 | 0.31 / 0.79 | 2.94 / 0.20 |
| PC+S+F | 1.68 / 7.12 | 1.63 / 8.64 | 0.53 / 1.09 | 6.64 / 1.85 | 80.44 / 75.10 | 8.19 / 5.47 | 0.89 / 0.73 |
| PC+D+F | 6.34 / 10.43 | 0.50 / 4.03 | 0.12 / 0.07 | 11.53 / 29.70 | 74.62 / 52.92 | 2.80 / 1.50 | 4.08 / 1.35 |
| **VGG7** | | | | | | | |
| BP | 25.94 / 3.73 | – / – | – / – | – / – | 61.56 / 92.30 | – / – | 12.49 / 3.97 |
| PC | 15.65 / 9.12 | 0.57 / 1.62 | – / – | 54.54 / 68.17 | 23.56 / 10.75 | 4.35 / 7.14 | 1.33 / 3.20 |
| iPC | 1.68 / 3.26 | 1.63 / 0.64 | – / – | 6.64 / 1.87 | 80.44 / 57.30 | 8.19 / 36.54 | 0.89 / 0.40 |
| iPC+S | 1.68 / 13.47 | 1.63 / 0.37 | 0.89 / 5.47 | 6.64 / 9.96 | 80.44 / 61.09 | 8.19 / 4.30 | 0.89 / 5.34 |
| PC+S+F | 3.30 / 2.38 | 0.28 / 5.03 | 4.13 / 0.52 | 10.29 / 7.41 | 54.20 / 79.25 | 1.80 / 3.43 | 25.99 / 1.97 |
| PC+D+F | 14.78 / 6.98 | 3.05 / 0.49 | 0.01 / 0.03 | 9.00 / 0.43 | 67.45 / 86.89 | 1.17 / 0.02 | 4.54 / 5.17 |
| **VGG10** | | | | | | | |
| BP | 9.69 / 1.88 | – / – | – / – | – / – | 87.86 / 98.10 | – / – | 2.45 / 0.02 |
| PC | 48.10 / 22.79 | 5.62 / 3.60 | – / – | 4.02 / 6.17 | 16.15 / 62.83 | 25.32 / 3.88 | 0.79 / 0.74 |
| iPC | 0.08 / 6.01 | 0.48 / 2.51 | – / – | 34.55 / 4.48 | 58.50 / 40.86 | 4.73 / 43.68 | 1.66 / 2.47 |
| iPC+S | 4.85 / 4.12 | 1.80 / 0.04 | 1.72 / 6.89 | 15.15 / 5.75 | 67.61 / 77.72 | 5.64 / 3.99 | 3.22 / 1.50 |
| PC+S+F | 4.84 / 3.76 | 1.55 / 0.40 | 2.08 / 1.25 | 9.80 / 6.71 | 77.14 / 82.10 | 2.16 / 3.04 | 2.43 / 2.74 |
| PC+D+F | 8.25 / 1.34 | 2.79 / 0.39 | 0.04 / 1.70 | 1.74 / 4.41 | 75.95 / 79.06 | 9.01 / 3.24 | 2.22 / 9.87 |
| **ResNet10** | | | | | | | |
| BP | 6.60 / 8.67 | – / – | – / – | – / – | 93.30 / 90.23 | – / – | 0.10 / 1.10 |
| PC | 8.87 / 20.05 | 0.46 / 4.46 | – / – | 30.41 / 10.44 | 33.60 / 46.35 | 21.77 / 14.47 | 4.89 / 4.23 |
| iPC | 1.47 / 11.93 | 1.31 / 3.77 | – / – | 19.18 / 9.98 | 61.87 / 59.41 | 2.87 / 13.44 | 13.30 / 1.47 |
| iPC+S | 4.74 / 0.32 | 0.22 / 2.10 | 1.07 / 6.33 | 11.68 / 21.00 | 59.33 / 44.40 | 11.79 / 1.57 | 11.16 / 24.28 |
| PC+S+F | 7.86 / 1.93 | 2.04 / 0.53 | 17.93 / 1.23 | 9.63 / 23.06 | 57.00 / 66.37 | 4.32 / 5.77 | 1.22 / 1.12 |
| **ResNet18** | | | | | | | |
| BP | 7.80 / 1.49 | – / – | – / – | – / – | 92.00 / 98.21 | – / – | 0.20 / 0.30 |
| PC | 11.39 / 0.84 | 2.86 / 4.17 | – / – | 55.11 / 24.00 | 12.75 / 42.24 | 8.48 / 7.79 | 9.40 / 20.96 |
| iPC | 3.76 / 2.37 | 1.57 / 7.89 | – / – | 14.85 / 13.94 | 47.67 / 19.73 | 11.08 / 9.87 | 21.07 / 46.20 |
| iPC+S | 5.82 / 1.20 | 0.55 / 0.42 | 1.95 / 0.78 | 2.95 / 52.00 | 78.29 / 41.70 | 8.83 / 1.90 | 1.61 / 1.99 |
| PC+S+F | 1.73 / 1.30 | 17.97 / 0.46 | 5.98 / 0.79 | 16.95 / 14.49 | 48.95 / 34.98 | 3.05 / 45.45 | 5.36 / 2.54 |

### I.2. Hyperparameter Transferability

To evaluate if hyperparameter searching is required for each setting, we conducted two sets of experiments to evaluate hyperparameter transferability.

**Across Datasets:** We took the optimal hyperparameters found on CIFAR-10 and applied them to CIFAR-100, and vice versa, for the VGG7 architecture.

**Across Architectures:** We took the optimal hyperparameters from VGG5 and VGG7 and applied them to the VGG10 model on CIFAR-10. Since the hyperparameter T needs to be larger than the number of layers, the T we used in these experiments is $max(10, T_{optimal})$.

The results, presented in Table 23 and 24, suggest that while optimal performance requires dedicated tuning, the hyperparameters show a reasonable degree of transferability, especially for our proposed methods. This indicates they are not pathologically sensitive to the specific dataset or architecture.

*Table 23.* Test Accuracies from Hyperparameters (HPs) Transfer Across Datasets on VGG7.

| Method | On CIFAR-10 | | On CIFAR-100 | |
|---|---|---|---|---|
| | Optimal CIFAR-10 HPs | Optimal CIFAR-100 HPs | Optimal CIFAR-100 HPs | Optimal CIFAR-10 HPs |
| BP | $89.91^{\pm0.12}$ | $88.75^{\pm0.07}$ | $65.36^{\pm0.15}$ | $65.23^{\pm0.24}$ |
| PC | $84.62^{\pm0.10}$ | $78.38^{\pm0.17}$ | $56.80^{\pm0.14}$ | $51.65^{\pm0.28}$ |
| S+F | $90.08^{\pm0.21}$ | $89.54^{\pm0.09}$ | $66.34^{\pm0.22}$ | $63.66^{\pm0.06}$ |
| iPC | $80.15^{\pm0.18}$ | $74.35^{\pm0.90}$ | $43.99^{\pm0.30}$ | $37.26^{\pm0.07}$ |
| iPC+S | $90.25^{\pm0.06}$ | $90.14^{\pm0.20}$ | $65.76^{\pm0.12}$ | $61.56^{\pm1.56}$ |

*Table 24.* Test Accuracies from Hyperparameters (HPs) Transfer Across Architectures (on CIFAR-10).

| Method | VGG10 with Optimal HPs | VGG10 with VGG7 HPs | VGG10 with VGG5 HPs |
|---|---|---|---|
| BP | $92.21^{\pm0.08}$ | $90.70^{\pm0.14}$ | $90.67^{\pm0.22}$ |
| PC | $72.75^{\pm6.03}$ | $79.20^{\pm0.19}$ | $49.66^{\pm0.54}$ |
| S+F | $92.07^{\pm0.10}$ | $88.95^{\pm0.18}$ | $90.67^{\pm0.11}$ |
| iPC | $63.83^{\pm0.33}$ | $72.30^{\pm0.55}$ | $73.78^{\pm0.15}$ |
| iPC+S | $93.03^{\pm0.18}$ | $90.60^{\pm0.60}$ | $92.29^{\pm0.13}$ |

