# OpenReview forum: "Towards the Training of Deeper Predictive Coding Neural Networks"
_ICML.cc/2026/Conference — ICML 2026 regular_

### Official Review · Reviewer_GL1E · 2026-03-11

**Soundness:** 3
**Presentation:** 3
**Significance:** 2
**Originality:** 2
**Overall Recommendation:** 3
**Confidence:** 3

**Summary:**

This paper studies why predictive coding networks break down at depth. The authors argue that the main issue is layer-wise energy imbalance: deeper layers accumulate much larger errors, while early layers receive weak learning signals. They address this with dynamic precision weighting, a forward-augmented weight update, BatchNorm Freezing, and auxiliary neurons for residual paths. These changes substantially improve deep PCNs and make VGG and ResNet models perform close to backpropagation on several image benchmarks.

**Compliance With Llm Reviewing Policy:**

Affirmed.

**Final Justification:**

The paper tackles an important problem and shows meaningful gains for deep predictive coding. The rebuttal clarified the VGG/ResNet distinction, component roles, and the non-essential role of the forward update, which improved my view. Still, method complexity and limited scaling evidence keep me from supporting acceptance.

**Key Questions For Authors:**

**Q0** The architectural distinction between the deep VGG experiments and the residual-network experiments is not fully clear. Once shortcut connections are introduced, the model is no longer a plain VGG-style PCNN, so the paper would benefit from a more explicit description of how the ResNet variant is constructed and how it differs from a residualized VGG backbone.

**Q1** The ablations are helpful, but the final method still combines several moving parts, so a clearer discussion of which components are essential in which settings would strengthen the paper.

**Q2** The role of the forward update deserves to be discussed more carefully because it appears important for performance but also makes the learning rule less biologically plausible.

**Q3** The paper would benefit from a more grounded discussion of how far the current method is expected to scale beyond the benchmarks used here, especially toward full-resolution ImageNet.

**Q4** Figure 2 provides strong evidence that energy imbalance is associated with the failure of deep PCNNs, but it does not by itself establish that this imbalance is the primary cause rather than one consequence of broader optimization difficulties.

**Limitations:**

yes

**Strengths And Weaknesses:**

**Strengths**
* the paper tackles an important problem since predictive coding methods work reasonably well in shallow networks but degrade badly in deep ones.
* The core intuition is strong and useful. The energy imbalance view gives a clear reason for why depth hurts training.
* The proposed methods are well connected to the defined problems. The fixes are not arbitrary and each one targets a specific failure mode.
* The experimental gains are meaningful. The new variants largely remove the collapse seen in standard PC and iPC as depth increases.
* The ResNet results are especially convincing because skip connections are one of the hardest cases for predictive coding.
* The paper is fairly honest about its own limitations, especially the reduced biological plausibility of the forward update.

**Weaknesses**
* The method is quite complex. The final recipe combines several changes, so it is hard to tell which part is truly necessary.
* The forward update improves accuracy, but it weakens the biological motivation because it depends on storing information from initialization.
* Even though the paper introduces a fix for residual networks, much of the evaluation still focuses on VGG-style models rather than a broader set of ResNet variants.
* The experiments are strong within this setting, but they do not yet show scalability to full-resolution ImageNet or similarly large tasks.

---

> ### Author Rebuttal · Authors · 2026-03-31
>
> We thank the reviewer for the time and effort.
>
> > Q0: The architectural distinction between the deep VGG experiments and the residual-network experiments is not fully clear. How is the ResNet variant constructed and how does it differ from a residualized VGG backbone?
>
> Our VGG models are plain sequential convolutional networks without any skip connections. Our ResNet variants follow the standard ResNet topology proposed in the original paper with explicit residual blocks and skip connections. We kept these architectures separate to isolate two distinct failure modes: the VGG experiments isolate the pure effect of depth on iterative signal attenuation. The ResNet experiments then isolate the effect of skip connections. Layer-by-layer details are in Appendix D (Table 4).
>
> > Q1: A clearer discussion of which components are essential in which settings would strengthen the paper.
>
> We ran a full factorial ablation (also discussed in our response 2 to Reviewer CK4G, to be added as a Table (The same as Figure 8). We will include a full factorial ablation table and a radar chart ([link to the figure](https://anonymous.4open.science/r/ICML_15242/fig_radar.pdf)) in the revision to show component-wise contributions:
>
> - Algorithmic updates (S and F) matter most for deep sequential models. On VGG, the contribution of S and F grows with depth.
>
> - Spiking Precision is surprisingly important for residual networks. Comparing VGG10 and ResNet-10 (both 10 layers, one sequential, one with skip connections), S is crucial for ResNets because skip connections create a learning rate ($\alpha$) conflict in standard PC: a large $\alpha$ helps sequential blocks but destabilizes skip paths. S resolves this by dynamically modulating the effective gain. The detailed reason will be included in the revision.
>
> - BF: Essential for PC to prevent overfitting of running statistics during iterative relaxation (Appendix F.4).
>
> - Auxiliary Neurons: Resolve temporal mismatches in skip-connections (Section 4.3.1).
>
> Our results on CIFAR100 vs CIFAR10 show that combining all methods becomes increasingly necessary on harder datasets.
>
> We will include these discussions in the revised manuscript and link each component to the extended analysis in the Appendix.
>
> > Q2: The role of the forward update deserves to be discussed more carefully because it appears important for performance but also makes the learning rule less biologically plausible.
>
> You are right that this deserves more discussion: we have decided to add our results on forward updates despite its implausibility as it points towards a problem of deep models: the neurons tend to converge to a value far away from that of the forward pass. However, we do not believe it to be a (strong) limitation of our work, as the best results are obtained without it. Hence, the claim that we manage to scale PC to large scale tasks while keeping the operations local in space and time still holds.
>
> > Q3: The paper would benefit from a more grounded discussion of how far the current method is expected to scale beyond the benchmarks used here, especially toward full-resolution ImageNet.
>
> Scaling to ImageNet-1K at $224 \times 224$ is currently constrained by compute, as our framework requires $\mathcal{O}(T \times L)$ sequential operations, making hyperparameter search extremely long. Instead, we demonstrated scalability across three axes: (1) depth (101-layer ResMLP), (2) label complexity (ImageNet32), and (3) resolution and complexity (Galaxy10 at $112 \times 112$). In all cases, we match BP within ~1%. We view scaling to full ImageNet as an engineering challenge (e.g., distributed simulation, optimized parallel relaxation) rather than a fundamental algorithmic limitation, and we will discuss this in the revision.
>
> > Q4: Figure 2 provides strong evidence that energy imbalance is associated with the failure of deep PCNNs, but does it establish that this imbalance is the primary cause rather than one consequence of broader optimization difficulties?
>
> We think energy imbalance is the primary cause, not a symptom, using theory and controlled iPC experiments:
>
> 1.  Theoretical Foundation: Theorem 1 proves top-down error decays at $\mathcal{O}(\alpha^{L-l})$ due to the small inference learning rate $\alpha$. This occurs at initialization, independent of weight quality.
> 2.  Isolating the Cause: iPC largely avoids representation drift. Yet, baseline iPC still collapses with depth. This isolates signal attenuation as the structural bottleneck.
> 3.  Targeted Intervention: If imbalance were a mere symptom, a targeted error-signal fix wouldn't work. However, applying Spiking Precision (iPC+S), which strictly restores energy distribution—dramatically recovers performance (e.g., VGG10 on CIFAR10 jumps from 63.83% to 93.03%). Figure 5 also visually confirms iPC+S successfully increases energy reaching the first layer from $10^{-6}$ to $10^{-3}$, directly driving this recovery.
>
> We will add this argument to Section 4.2.1.

---

> > ### Author Rebuttal · Reviewer_GL1E · 2026-04-05
> >
> > Thank you for the rebuttal. It addresses several of my main questions and makes the paper clearer. I especially appreciate the clarification of the distinction between the VGG and ResNet settings, and the added explanation of how different components address different failure modes. The planned factorial ablation should also make the final version easier to interpret.
> >
> > The response on the forward update and on energy imbalance is also helpful. I appreciate the clarification that the forward update is not necessary for the best-performing version, and the added argument makes the interpretation of energy imbalance more convincing, even if I still think causality should be stated carefully.
> >
> > Some limitations remain, especially the method’s complexity and the still-limited evidence for scaling to larger settings. But overall, the rebuttal resolves several of my main concerns and improves my view of the paper.

---

> > > ### Author Response · Authors · 2026-04-06
> > >
> > > Thank you for your comment and positive feedback. We are incorporating the outcomes of the discussions with you and other reviewers in the manuscript, which we believe have largely improved the quality and clarity of the submission.
> > >
> > > As you stated that our rebuttal and further experiments have improved your view of the paper, do you think it would be possible to also update the score to reflect this?

---

### Official Review · Reviewer_bFhK · 2026-03-11

**Soundness:** 4
**Presentation:** 3
**Significance:** 3
**Originality:** 3
**Overall Recommendation:** 5
**Confidence:** 3

**Summary:**

This study proposes multiple technical improvements on predictive coding models, allowing them to scale to deeper networks and residual networks. The improvements proposed are: dynamic reweighting of layerwise losses via the precision term, a modification of the equilibrium loss term which anchors it to the initial forward path to avoid drifts, a delay line on residual connections to synchronize error propagation with the main branch, and an adaptation of batchnorm. With these improvements, they show that deeper networks such as ResNet-18 and VGG-10 trained via predictive coding can approach backpropagation accuracy on datasets such as TinyImageNet and Imagenet32 (low resolution Imagenet).

**Compliance With Llm Reviewing Policy:**

Affirmed.

**Final Justification:**

I maintain my positive score for this study which constitutes in my opinion a valuable addition to the predictive coding literature (see Strengths).

**Key Questions For Authors:**

Instead of spiking precision, have you tried to use a fixed but layer-specific precision? Does that not work as well?

**Limitations:**

yes

**Strengths And Weaknesses:**

Soundness: The submission is technically sound. The proposed modifications are well justified and demonstrated to work with adequate experiments. The claims are well aligned with the results of the study.

Presentation: The article is targeted at a rather specialized audience but it is clearly written and structured. Prior work is properly discussed and the innovations are clear.
l 256 "In terms of temporal scheduling, this happens at l = L − t." => t = L-l

Significance: The results appear to be quite significant. On TinyImagenet with a ResNet-18, their best method does 77.55% compared to 79.94% for backprop, and appear to substantially outperform previous predictive coding baselines (e.g 11.58%)

Originality: The paper provides interesting insights, in particular the diagnoses about why energy stays stuck in layers close to the top and don't propagate to lower layers in normal predictive coding, how the spiking precision improves on this, why normal resnets are not adapted to predictive coding because of asynchronous updates, etc.

---

> ### Author Rebuttal · Authors · 2026-03-30
>
> We thank the reviewer for the time and effort.
>
> > W1: Typo at l 256 "$l = L - t \Rightarrow t = L - l$"
>
> Thank you for mention this. We will correct this in the revised version.
>
> > Q1: Instead of spiking precision, have you tried to use a fixed but layer-specific precision? Does that not work as well?
>
> This is an insightful question, we had not tried this before your suggestion. We have now run the experiment.
>
> We set up a fixed, layer-specific precision that scales linearly across the network: bottom-layer precision $a$, top-layer precision $b$, with intermediate layers linearly interpolated. We treated $a$ and $b$ as hyperparameters and searched over them. The best configuration was $a = 0.9$ (bottom) and $b = 0.05$ (top), achieving $88.21\%$ on VGG10/CIFAR10. This is better than the PC+F baseline ($87.33\%$) because it partially mitigates the spatial energy imbalance, but below Spiking Precision ($92.07\%$).
>
> **Table: Comparison of test accuracy across different precision strategies on VGG10/CIFAR10.**
>
> | **Method** | **Test Accuracy** |
> | :--- | :--- |
> | PC+F (Baseline) | $87.33\%$ |
> | PC + Fixed Layer-specific Precision + F | $88.21\%$ |
> | PC + Spiking Precision + F | $\mathbf{92.07\%}$ |
>
> The fixed scheme helps, but Spiking Precision does substantially better because it also addresses the temporal aspect of the problem: it concentrates the gain at the exact moment the error signal first arrives at each layer, rather than applying a static rescaling.

---

> > ### Author Rebuttal · Reviewer_bFhK · 2026-04-03
> >
> > I thank the authors for the interesting additional experiment which I believe should be added to the study. I maintain my positive score for this study which constitutes a valuable addition to the predictive coding literature.

---

> > > ### Author Response · Authors · 2026-04-04
> > >
> > > We thank the reviewer for the positive feedback and agree that this experiment strengthens the paper; we will include it in the final version.
> > > We also sincerely appreciate the reviewer’s time and effort throughout the review process.

---

### Official Review · Reviewer_4FpB · 2026-03-11

**Soundness:** 4
**Presentation:** 3
**Significance:** 3
**Originality:** 3
**Overall Recommendation:** 4
**Confidence:** 2

**Summary:**

This paper proposes methods to address the difficulty of training deep models with predictive coding (PC). The authors hypothesize that a key reason PC fails to scale in depth is an inter-layer energy imbalance: in deep networks, the energy (prediction-error term) in early layers becomes extremely small compared to that in later layers near the output, which weakens learning signals for shallow layers. To mitigate this, they introduce two main techniques, Spiking Precision and Forward Updates.

First, Spiking Precision modifies the layer-wise precision/covariance during inference in a time-dependent manner: at the moment when the energy term reaches a given layer, the precision is briefly adjusted (scaled according to the inference update rate), effectively “boosting” the propagation of error signals from output to input and reducing the concentration of energy in later layers. Second, Forward Updates changes the weight-update formulation by using an error term defined with respect to the forward-pass prediction, aiming to reduce the divergence between the inferred latent states and their forward-pass values.

Empirically, these techniques improve training stability and test accuracy for VGG-style networks trained with both PC and incremental PC (iPC). The paper further considers architectures with skip connections. It attributes failures on ResNet-like models to a mismatch in the timing of energy propagation along shortcut paths, and proposes inserting auxiliary (dummy) nodes on residual connections to synchronize phases. In addition, the authors propose freezing BatchNorm statistics during inference. With these modifications, they demonstrate that PC and iPC can successfully train ResNet-18 in their experiments.

**Compliance With Llm Reviewing Policy:**

Affirmed.

**Final Justification:**

I keep my positive score because the authors adequately addressed my concerns.

**Key Questions For Authors:**

1. The paper mainly studies convolutional networks. Do the authors expect the proposed techniques to extend to attention-based architectures such as Transformers? If so, what changes would be required (e.g., for skip connections, normalization layers, or the inference dynamics)?
2. Is PC or iPC expected to be more efficient than backpropagation in practice? If not currently, do the authors envision settings (hardware, parallelization, amortization, etc.) where PC/iPC could become more energy-efficient or computationally efficient than backprop?
3. Many modern architectural choices (e.g., skip connections and batch normalization) were developed with backprop-based optimization in mind. The proposed fixes (auxiliary nodes for residual paths, BN statistics freezing) suggest that these architectures may not be immediately “PC-native.” If one were to design architectures specifically for PC/iPC training, do the authors expect additional gains in stability or efficiency? Are there architectural principles the authors would recommend for PC/iPC-friendly model design?

**Limitations:**

yes

**Strengths And Weaknesses:**

### Soundness

The paper empirically characterizes key failure modes of PC and iPC and proposes concrete remedies. The proposed techniques are validated on representative architectures (VGG and ResNet), and the experiments indicate that the methods substantially improve training stability and accuracy. Overall, the soundness of the approach appears reasonably supported by the experimental results.

### Presentation

The paper is well structured and easy to follow. The motivation, problem analysis, and proposed components are presented in an organized manner.

### Significance

If the proposed techniques indeed enable stable training of deeper models with PC/iPC where prior attempts struggled, this addresses an important bottleneck in making predictive-coding-based learning practical beyond shallow settings. In that sense, the paper tackles a meaningful problem for alternatives to standard backpropagation.

That said, I am slightly concerned about the motivation in the introduction: while the paper argues that deep learning is costly from an energy-consumption perspective, it provides limited discussion of the computational/energy cost of PC and iPC themselves. Since PC-style methods typically involve iterative inference, it is not obvious that they are algorithmically more efficient than backpropagation in standard training setups. A clearer discussion (or measurements) of wall-clock/compute/energy trade-offs would strengthen the motivation.

### Originality

I am not fully familiar with the recent literature on scaling predictive coding to deep architectures, so I am not completely confident about the novelty relative to concurrent work. However, the specific combination of analysis and proposed mechanisms (e.g., the precision scheduling and the modifications for residual connections and normalization) appears likely to contain original contributions.

---

> ### Author Rebuttal · Authors · 2026-03-31
>
> We thank the reviewer for the time and effort.
>
> > Limited discussion of the computational/energy cost of PC and iPC themselves. A clearer discussion/measurements of wall-clock/compute/energy trade-offs would strengthen the motivation.
>
> Thanks for your suggestion. Please see our response to Q2 below for wall-clock time comparisons. We will also add a brief discussion to the introduction clarifying that the efficiency argument for PC is about hardware compatibility (spatial/temporal locality enables analog neuromorphic implementations), not about beating BP on standard GPUs.
>
> > Q1: Do the authors expect the proposed techniques to extend to attention-based architectures such as Transformers? If so, what changes would be required?
>
> We expect the core principles to transfer, but with non-trivial adaptations.
>
> The energy imbalance is architecture-agnostic: error signals would still decay across many layers in a Transformer trained with PC, so Spiking Precision would be needed. The representation drift that motivates Forward Update would also persist, since the inference phase in a Transformer would likewise push neural activities away from their feed-forward values. For residual connections in Transformers, the phase mismatch we identified would also arise, so the auxiliary node technique would be directly applicable.
>
> The main open question is how to formulate an energy term for attention weights, since attention involves input-dependent routing rather than fixed linear projections. This is a direction we are actively exploring.
>
> > Q2: Is PC or iPC expected to be more efficient than backpropagation in practice? If not currently, do the authors envision settings where PC/iPC could become more energy-efficient?
>
> On standard GPUs, PC and iPC are not more efficient than BP. As shown in Appendix E (Table 7), the $T$ inference steps make PC slower: roughly $O(T \times L)$ per training step versus $O(L)$ for BP. On a single H100, training ResNet18 takes ~15s per epoch for PC versus ~2.8s for BP.
>
> The efficiency argument is about hardware, not algorithms. PC's updates are local in space and time, which means they can be mapped onto analog neuromorphic hardware (e.g., memristor crossbars) where the relaxation phase happens in continuous physical time rather than being simulated sequentially. On such hardware, the $O(T \times L)$ bottleneck disappears. We will state this more clearly in the revision.
>
> > Q3: If one were to design architectures specifically for PC/iPC training, do the authors expect additional gains? Are there architectural principles recommended for PC/iPC-friendly model design?
>
> Yes, we expect gains from PC-native design. Our results already point to two principles:
>
> First, normalization layers that rely on accumulating statistics over the inference phase (like BatchNorm) need special treatment. PC-native architectures might benefit more from local normalization schemes (like LayerNorm) that do not require freezing during inference.
>
> Second, architectures with strictly synchronized feedback loops would naturally avoid the temporal mismatches we had to fix with auxiliary neurons.

---

> > ### Author Rebuttal · Reviewer_4FpB · 2026-04-02
> >
> > Thank you for the detailed rebuttal.
> > I have read your responses carefully, and most of my concerns have been addressed.
> >
> > I will maintain my current score.

---

> > > ### Author Response · Authors · 2026-04-04
> > >
> > > We thank the reviewer for the careful reading of our rebuttal and for the time and effort devoted to evaluating our work; we respect the decision to maintain the score and are glad that our responses addressed most of the reviewer’s concerns.

---

### Official Review · Reviewer_CK4G · 2026-03-13

**Soundness:** 3
**Presentation:** 2
**Significance:** 2
**Originality:** 2
**Overall Recommendation:** 4
**Confidence:** 3

**Summary:**

The paper studies how to train deeper predictive coding networks and aims to bring their performance closer to that of backpropagation on modern deep models. The authors start from the observation that predictive coding can perform competitively in relatively shallow networks, but its performance degrades substantially as depth increases, particularly in architectures with residual connections. To address this, the paper proposes several modifications intended to improve training stability and performance in deeper settings.

The paper combines several different kinds of components. PC and iPC are learning or training frameworks rather than architectures, while VGG, ResNet, and ResMLP are the underlying network architectures. On top of these, the paper studies several additional mechanisms at different levels: Spiking Precision and Forward Update modify the training dynamics, auxiliary neurons are an architectural modification targeted at residual networks, BatchNorm Freezing changes the behavior of BatchNorm during training and inference, and center nudging is an existing supervision heuristic discussed alongside the proposed methods. The experiments therefore compare methods across both learning framework and architecture, while also evaluating the effects of these additional mechanisms.

The number of combinations is a bit dizzying but my general impression is that the best mechanism depends on the base learning framework, with standard predictive coding favoring Spiking Precision plus Forward Update, and incremental predictive coding favoring Spiking Precision alone.

**Compliance With Llm Reviewing Policy:**

Affirmed.

**Final Justification:**

I don't have a background in predictive coding or "deep" predictive coding so focused my engagement at a structural level. My main concerns might be summarised as "many moving parts", which I think GL1E also shared. I accept iterative engineering work is sometimes the work that is required to make things ultimately work. I raised my score from 3 to 4.

**Key Questions For Authors:**

1. Is there a strong reason for separating the ``contributions" into algorithmic and structural? I don't think BatchNorm Freezing fits so cleanly into structural.
2. The paper studies multiple kinds of modifications at once, including different learning frameworks (PC vs iPC), architecture choices, training-dynamics interventions, residual-network interventions, and BatchNorm-related changes. Could the authors clarify the experimental design more explicitly, ideally as a table of which components are active in each reported configuration, and explain why a more systematic factorial-style ablation was not used? As written, it is difficult to tell which gains should be attributed to individual interventions and which arise from interaction effects among them.

**Limitations:**

Yes

**Strengths And Weaknesses:**

Soundness:

The paper appears broadly technically sound in the sense that the proposed interventions are concrete, implementable modifications to predictive-coding systems, and the authors evaluate them empirically across multiple architectures and datasets. The submission also provides mathematical formulation for the main mechanisms and includes algorithmic detail and appendix material supporting the proposed methods. I did not find the work obviously unsound or internally inconsistent at a high level.

My main reservations are less about whether the interventions are legitimate and more about how cleanly the experiments isolate their effects. In particular, the paper studies multiple kinds of changes at once, including learning frameworks, architectures, training-dynamics modifications, residual-network modifications, and normalization behavior, which can make attribution of gains difficult. So my questions are mainly about experimental design and decomposition of effects rather than about basic technical validity.

Presentation:

I found the paper difficult to read. The exposition is jargon-heavy and often motivates the proposed mechanisms informally, whereas I would have preferred a cleaner mathematical statement of the failure mode, the assumptions, and why the proposed fixes should work.

In addition, while there is two baseline methods PC and iPC, and six interventions, this is not a full factorial design because not every combination is sensible. It becomes pretty hard to keep track.

Significance:

The contribution reads primarily as an engineering paper: the authors identify a practical issue in training deep predictive-coding networks, introduce a collection of interventions, and show empirically that performance improves. I can see the value of this for researchers working directly on deep predictive coding, but I found it less compelling as a conceptual or theoretical contribution.

Originality:

The paper proposes a diagnosis of the depth-scaling problem in predictive coding, rather than definitively establishing it. In particular, it argues that performance degradation is related to imbalanced error propagation across layers, drift between feed-forward and relaxed states, and timing mismatches in residual pathways. These hypotheses are supported by a mix of theoretical arguments, visualizations, and intervention-based experiments, but they do not amount to a definitive explanation of the underlying cause. (But that's ok, it's hard to get rigorous formal results for real deep learning...)

I suppose the work can be rated as original in that it designs ``original" interventions motivated by those degradation factors and show that said interventions help empirically.

---

> ### Author Rebuttal · Authors · 2026-03-30
>
> We thank the reviewer for the time and effort.
>
> > The contribution reads primarily as an engineering paper: the authors identify a practical issue in training deep predictive-coding networks, introduce a collection of interventions, and show empirically that performance improves.  I can see the value of this for researchers working directly on deep predictive coding, but I found it less compelling as a conceptual or theoretical contribution.
>
> Scaling up learning algorithms, especially bio-plausible ones, requires a significant engineering effort, as you have to constantly track how all the quantities (energies, spectral norms, weights, accuracies, etc.) evolve during training, to make sure  the dynamical system they are part of is stable. Doing so, and find ways of addressing any problem that arises is vital when it comes to scale towards larger scale tasks. You stated that you do see the value "for people working directly on predictive coding": creating value in that research community is the goal of our work. About the limited scope outside of the predictive coding literature, we  believe that the general engineering 'reasoning' outlined in this answer (1. observe quantities that make your dynamical system unstable, 2. find a way to address the cause which does not alter bio-plausibility, 3. check if it works at scale, 4. repeat) can be of interest to researchers solving the scalability problem of other biologically plausible learning algorithms.
>
> > Reason for separating the “contributions” into algorithmic and structural.
>
> Our initial rationale was that Spiking Precision and Forward Update are architecture agnostic, as they simply modify the learning rules of PC, while Residual Connections and BatchNorm Freezing involve changes to how network layers behave during the inference phase. From a pytorch perspective (as an example), the first require changes to the 'optimizer' or 'nodes', while the second would require changes to nn.Modules. But we agree that BF as a structural change is misleading. In the revision, we have re-organized the contributions along functional lines: *Algorithmic Contributions* (Spiking Precision, Forward Update), and *Architecture Adaptation* (Residual Connections, BatchNorm Freezing), and better motivated this.
>
> > My main reservations are less about whether the interventions are legitimate and more about how cleanly the experiments isolate their effects.
>
> We'd like to point out that an initial, factorial-style ablation with a large number of experiments highlighting how different combinations influence performance and stability (energy decay) is present in the appendix F. In addition to that, we have performed a second, large study where we test combinations in isolation. For every combination of configurations, we ran a comprehensive hyperparameter search, analyzed the test accuracies, and plotted how well each combination mitigates 'energy decay'. After your suggestion we have now further improved it (as described below), and also refer to it more in detail from the main body. Please let us know if there is any kind of further factorial ablation you'd like us to run. More in detail:
>
> Auxiliary Neurons are Isolated in Figure 4(b) of the main text.
>
> S, F, and BF: Table 15 in the appendix provides a factorial breakdown of S and F. Figure 8 extends this to all three components (S, F, BF), testing each alone, in pairs, and combined. We presented these as a plot to show the stabilizing effect of BF visually, but we now see that this obscured the systematic nature of the study. We will also add this unified ablation table (Anonymous and secure link: ([link to the table](https://anonymous.4open.science/r/ICML_15242/Ablation_study_results.pdf)) ) to the main text. A more detailed analysis can be found in our response to Q1 from Reviewer GL1E.
>
> Thank you for the insight: we have adapted the text to better specify the advantages of the individual changes in isolation, and complemented this via  references to the individual sections in the supplementary material. In addition, we have added the tables linked above. Let us know whether this answer and provided evidence is satisfactory for you, or whether you believe that a specific experiment/ablation study that would better isolate the effects of a single method is missing. If that is the case, please point it out to us, and we will be happy to run it quickly, and report here the results.

---

> > ### Author Rebuttal · Reviewer_CK4G · 2026-04-03
> >
> > Thank you for addressing my concerns and for the thoughtful responses to the other reviewers. The additional ablation table and reorganisation of contributions clarify the experimental design. I am raising my score from 3 to 4.

---

> > > ### Author Response · Authors · 2026-04-04
> > >
> > > We thank the reviewer for the encouraging feedback and for raising the score. We appreciate the suggestions on the ablation study table and the reorganization of contributions, which have improved the clarity and overall quality of the paper and will be incorporated in the final version. Thank you once again for your time and effort.
> > >
> > > P.S.: Would it be possible to officially raise the score to 4? The official displayed score is still a 3.

---

### Decision · Program_Chairs · 2026-04-30

**Decision:**

Accept (regular)

**Comment:**

This paper studies the causes behind the poor performance of predictive coding-based deep networks. It identifies a number of techniques to facilitate training that ensure that the prediction-error at different layers is roughly balanced. Reviewer 4FpB primarily had clarification questions which have been adequately addressed by the rebuttal. The authors also conducted a new experiment while responding to comments by Reviewer bFhK regarding fixed, layer-specific precision. The two main concerns that could not be addressed sufficiently where as follows. Multiple reviewers (CK4G and GL1E) had concerns about the complexity of the method and the different moving parts. Reviewer GL1E also had concerns about the limited scalability of the proposed training method. The authors have provided an ablation study to argue that the different components (precision schedule and forward updates, spiking precision, batch-norm freezing, auxiliary neurons) are indeed necessary for this approach.

This paper has developed a keen insight into a very interesting technical question. Although the current approach is a bit complicated, it is a great step forward. And there is sufficient evidence in the paper that the approach works in practice. I am happy to recommend that the paper be accepted.